# ATAD2 mediates chromatin-bound histone chaperone turnover

**Ariadni Liakopoulou[1], Fayçal Boussouar[1]\*, Daniel Perazza[1], Sophie Barral[1], Emeline Lambert[1], Tao Wang[1], Florent Chuffart[1], Ekaterina Bourova-Flin[1], Charlyne Gard[2], Denis Puthier[2], Sophie Rousseaux[1], Christophe Arnoult[1], André Verdel[1], Saadi Khochbin[1]\***

[1]Université Grenoble Alpes, INSERM U1209, CNRS UMR 5309, Institute for Advanced Biosciences, Grenoble, France; [2]Aix-Marseille Université, Marseille, France

## eLife Assessment

This **valuable** study explores the role of the chromatin regulator ATAD2 in mouse spermatogenesis. The data **convincingly** demonstrate that ATAD2 is essential for proper chromatin remodeling in haploid spermatids, influencing gene accessibility, H3.3-mediated transcription, and histone eviction. Using Atad2 knockout (KO) mice, the authors link ATAD2 to the DNA-replication-independent incorporation of sperm-specific proteins like protamines and histone H3.3. Although the findings highlight chromatin abnormalities and impaired in vitro fertilization in KO mice, natural fertility remains unaffected, suggesting possible in vivo compensatory mechanisms. Future experiments will be needed to tease out the precise molecular role of ATAD2 in spermatogenesis. This work will be of interest to the epigenetics and developmental fields.

**\*For correspondence:**
faycal.boussouar@univ-grenoble-alpes.fr (FB);
saadi.khochbin@univ-grenoble-alpes.fr (SK)

**Competing interest:** The authors declare that no competing interests exist.

**Abstract** ATAD2, a conserved protein which is predominantly expressed in embryonic stem (ES) cells and spermatogenic cells, emerges as a crucial regulator of chromatin plasticity. Our previous parallel studies conducted in both ES cells and *S. pombe* highlighted the fundamental role of ATAD2 in facilitating chromatin-bound histone chaperone turnover. Focusing on mouse spermatogenesis, we demonstrate here that ATAD2 regulates the HIRA-dependent localization of H3.3 on the genome and influences H3.3-mediated gene transcription. Moreover, by modulating histone eviction and the assembly of protamines, ATAD2 ensures proper chromatin condensation and genome packaging in mature sperm. Disruption of *Atad2* function in mice leads to abnormal genome organization in mature spermatozoa. Together, these findings establish a previously overlooked level of chromatin dynamic regulation, governed by ATAD2-controlled histone chaperones binding to chromatin, which defines the balance between histone deposition and removal.

## Introduction

Spermatogenesis involves unique and dramatic chromatin remodeling and genome reorganization events that take place in preparation for a shift from the universal nucleosome-based eukaryotic genome organization into a unique nucleoprotamine-based structure (*Gaucher et al., 2010*; *Bao and Bedford, 2016*; *Le Blévec et al., 2020*; *Okada, 2022*). During the last decade, important information on the molecular basis of the nucleosome-to-nucleoprotamine transition during the postmeiotic phases of spermatogenesis became available. Several testis-specific H2A, H2B, and H3 histone variants replace the canonical histones, locally or globally, at different stages, starting early during the commitment of cells into meiotic and post-meiotic differentiation (*Hoghoughi et al., 2018*). Most of this nucleosome dismantling and assembly occurs in the absence of DNA replication.

In other words, the replication-independent nucleosome assembly pathways, such as the well-documented HIRA-dependent H3.3 assembly, are expected to be critical in ensuring the necessary chromatin remodeling in these cells. Finally, hints to the functions of transition proteins (TPs) and their functional relationship with protamines (PRMs) were proposed based on the study of a specific late-expressing H2A variant, H2A.L.2 (*Barral et al., 2017b*). In the proposed model, we suggest that TPs do not directly replace histones, but rather could be loaded on open H2A.L.2-containing nucleosomes to control PRM assembly. Nucleosome-to-nucleoprotamine transition also includes a TP-dependent pre-PRM2 processing, histone displacement by PRMs and the structuration of the nucleoprotamine-based genome organization in the mature spermatozoa (*Barral et al., 2017b*; *Rezaei-Gazik et al., 2022*). In mice, two protamines, PRM1 and PRM2, replace histones. PRM2 is initially synthesized as a precursor protein, which undergoes sequential proteolytic processing at its N-terminal region during its incorporation, ultimately producing the mature, fully processed PRM2 (*Arévalo et al., 2022*).

Taking into account this unique dynamic of post-meiotic cells' nucleosomes, it is expected that these cells massively express generic factors, or those predominantly expressed in spermatogenic cells that act on chromatin, including histone acetylating machineries, acetylated histone readers, and histone variants and their chaperones, all of which mediate histone and non-histone protein exchanges. More specifically, the replication-independent nature of nucleosome assembly, both in cells undergoing meiosis and in post-meiotic cells, strongly suggests the involvement of the chaperone/histone substrate couple, HIRA and H3.3.

An important part of this replication-independent histone assembly and transitions in the genome organizations occurs in the context of a massive histone H4 hyperacetylation in the haploid spermatogenic cells, spermatids, mediated by NUT-p300/CBP histone acetyltransferase complex (*Shiota et al., 2018*; *Rousseaux et al., 2022*). Hyperacetylation of histone H4, especially the double acetylation on H4K5K8 in these cells, leads to the binding of H4 by the first bromodomain of BRDT and to histone replacement (*Morinière et al., 2009*; *Gaucher et al., 2012*; *Goudarzi et al., 2016*; *Shiota et al., 2018*). Additionally, histone H4 hyperacetylation, especially at K5, could also be recognized by another bromodomain-containing factor of unknown function in spermatogenic cells, ATAD2 (*Caron et al., 2010*; *Morozumi et al., 2016*).

ATAD2 is a highly conserved bromodomain-containing factor that also harbors an AAA+ATPase domain (*Boussouar et al., 2013*; *Cattaneo et al., 2014*). Our early work showed that *Atad2* is predominantly expressed in ES cells, as well as in spermatogenic cells, and is aberrantly activated in virtually all solid cancers (*Caron et al., 2010*). Although ATAD2's function in cancer prompted many investigations (*Liu et al., 2022*), its function in its physiological context of expression has remained largely overlooked. We reported the first investigations of ATAD2's function in ES cells (*Morozumi et al., 2016*). These studies in ES cells showed that ATAD2 is recruited into acetylated transcriptionally active chromatin and is critical to keep the chromatin of ES cells highly dynamic, as illustrated by an ATAD2-dependent high rate of histone exchange (*Morozumi et al., 2016*). Subsequent studies conducted by our laboratory, in ES cells and in *S. pombe*, deciphered the molecular basis of this activity of ATAD2, which relies on its capacity to ensure the turnover of the chromatin-bound histone chaperone, HIRA and FACT (*Wang et al., 2021*). In addition to ES cells, *Atad2* is also highly expressed in male germ cells, pointing to its potential role in the regulation of HIRA-dependent H3.3 assembly in these cells, especially in post-meiotic cells. Indeed, previous published works have highlighted the important function of H3.3 assembly in meiotic and post-meiotic male genome programming (*van der Heijden et al., 2007*; *Yuen et al., 2014*; *Tang et al., 2015*; *Fontaine et al., 2022*). We therefore hypothesized that ATAD2-regulated HIRA-H3.3 activity would also be a critical player in the post-meiotic male genome reorganization, as HIRA has been shown to be essential post-fertilization (*Lin et al., 2014*) and during oogenesis (*Nashun et al., 2015*).

Using a lacZ tagged *Atad2* knockout (KO) mouse model, we could show that *Atad2*'s expression is enhanced in post-meiotic male germ cells, ensuring a proper turnover of histone H3.3, including on the sex chromosomes. More remarkably, our data demonstrate that ATAD2 controls the appropriate expression of a series of genes whose activity is known to be H3.3-dependent in post-meiotic cells. Additionally, in the absence of ATAD2, perturbed nucleosome structural organization leads to a delayed histone-to-PRM replacement, leading to a defective global final male genome compaction. Although this defect in mature spermatozoon genome compaction has minimal effect on male fertility

during natural reproduction, it impacts the success rate of in vitro fertilization using spermatozoa from *Atad2* KO mice, pointing to a more fragile genome in these cells.

In summary, our work revealed a novel mechanism for the regulation of chromatin plasticity, involving the control of histone chaperone actions, particularly that of HIRA, controlling the proper male germ cell transcriptional regulation, genome compaction, and optimal male fertility.

## Results
### Atad2 is predominantly expressed in haploid male germ cells

*Atad2* is normally expressed in embryonic stem (ES) and spermatogenic cells and remains silent in the majority of adult tissues (*Caron et al., 2010*; *Morozumi et al., 2016*). Therefore, in addition to ES cells, male germ cells constitute the second cell type where the molecular basis of the physiological ATAD2 activity could be investigated.

To decipher the function of ATAD2 during spermatogenesis, we used a mouse model harboring an insertion of a *lacZ* reporter cassette between exons 11 and 12 that enables splicing and fusion of the *lacZ* transgene with exon 11 of the long and short forms of *Atad2* (*Figure 1A*) and production of a non-functional ATAD2-lacZ fusion protein (*Figure 1B*), referred to as *Atad2* KO in this study. Furthermore, we crossed these mice with transgenic mice expressing ubiquitously Cre recombinase under *CMV* promoter, in order to remove the *neo* cassette and to delete *Atad2's* exon 12 (*Figure 1B*). In addition to the long somatic form of ATAD2, ATAD2L, the mouse testis encodes a short isoform with predominant expression in spermatogenic cells that we previously named ATAD2S (*Caron et al., 2010*). The combined LacZ fusion and *Cre recombinase*-dependent deletion of *Atad2's* exon 12 makes both ATAD2L and ATAD2S indetectable with our homemade antibody, while, as expected, a reduced level of both ATAD2 isoforms in *Atad2* heterozygous cells can be observed (*Figure 1C*).

The expression of *lacZ* under the control of the *Atad2* promoter allows one to visualize *Atad2* gene activity in a stage-specific manner. X-gal staining of seminiferous tubule sections from *Atad2* KO mice showed a remarkably high expression of *Atad2* in post-meiotic spermatogenic cells (*Figure 1D*). It is noteworthy that, as expected, no staining was observed in the wild-type tubules.

### ATAD2 controls HIRA accumulation

In our previous work performed in both *S. pombe* and mouse ES cells, we demonstrated the crucial role for ATAD2 in HIRA-dependent histone dosage as well as in HIRA-chromatin interaction. Among other discoveries, this work showed that in the absence of ATAD2, HIRA becomes trapped on chromatin leading to its accumulation in ES cells, as well as in a variety of cancer cell lines (*Wang et al., 2021*). Because of this very conserved action of ATAD2 in controlling the HIRA activity, we reasoned that the ATAD2-dependent HIRA activity should be particularly active during spermatogenic cell differentiation, especially in post-meiotic cells, when ATAD2 expression is highly induced (*Figure 1D*). To test if the absence of ATAD2 could lead to the accumulation of HIRA, as we previously reported in other systems, we quantified HIRA in spermatogenic cells from both wild-type and *Atad2* KO mice. These investigations revealed an accumulation of HIRA protein in *Atad2* KO spermatogenic cells (*Figure 2A*), consistent with its increased association with chromatin (*Wang et al., 2021*). We have also monitored HIRA accumulation as a function of spermatogenic cell stages. Due to the strong activation of *Atad2* after meiosis in spermatids (*Figure 1D*), the ATAD2-dependent accumulation of HIRA was expected to mostly occur in post-meiotic cells. Accordingly, we also compared HIRA accumulation in fractionated spermatogenic cell extracts. This experiment demonstrated that, in the absence of functional ATAD2, the accumulation of HIRA primarily occurs in round spermatids (*Figure 2B*). This finding is consistent with the post-meiotic activation of Atad2 and its role in regulating the turnover of chromatin-bound HIRA.

### ATAD2 controls HIRA-dependent H3.3 localization

We speculated that an increased presence of HIRA on chromatin in the absence of ATAD2 should also be associated with increased amounts of deposited histones. Indeed, we have demonstrated previously in *S. pombe* that histone gene inactivation could rescue the growth arrest due to the absence of *Atad2/Abo1*. In parallel, we found that in ES cells, in the absence of ATAD2, an increased nucleosome assembly occurs, even on the so-called nucleosome-free region (NFR) (*Wang et al., 2021*).

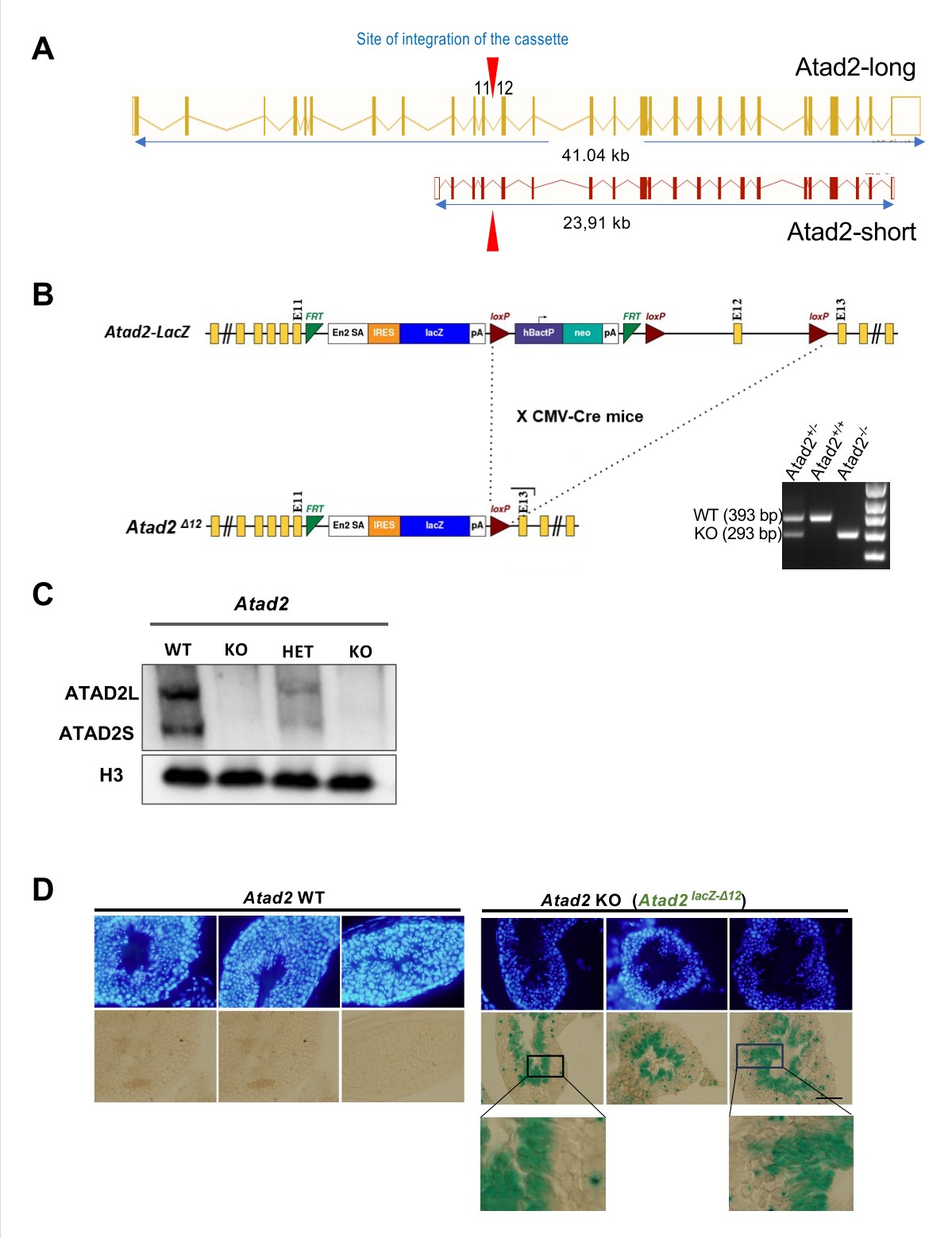

**Figure 1.** *Atad2* is highly expressed in post-meiotic male germ cells. (**A**) The exon/intron organization of *Atad2* gene and the two ATAD2-encoding transcriptional units (encoding for ATAD2-long and ATAD2-short) are indicated (***Dyer et al., 2025***). The red arrows indicate the position of insertion of the *Lac-Z-Lox-Neo-Lox* cassette. (**B**) Gene targeting strategy for the generation of the *Atad2* KO allele is represented. Exon number and relative position, selection gene cassettes, LoxP sites, and Frt sites are indicated. Crossing mice bearing this construct with *CMV-Cre* mice resulted in the generation of the *Atad2 Lac Z^{ΔNeo-Exon12}* allele, which can be verified through PCR amplification of the genomic DNA (shown on the ethidium bromide-

*Figure 1 continued on next page*

*Figure 1 continued*

stained gel). The expected amplified bands for each genotype allowing the detection of both wild-type and KO alleles of *Atad2* are indicated. E=Exon. (**C**) Total protein extracts from wild-type (WT), heterozygous (HET) and *Atad2* homozygous knockout (KO) mice testes were probed with anti-ATAD2 and anti-H3 antibodies as indicated. Four different mice were used for this experiment. (**D**) Seminiferous tubule sections from wild-type and *Atad2* KO mice were stained with X-gal to visualize *lacZ* gene expression and β -galactosidase activity under the endogenous *Atad2* gene promoter. Testes from three different wild-type and *Atad2* KO mice were used to generate the represented sections. Scale bar: 200 μm.

The online version of this article includes the following source data for figure 1:

**Source data 1.** PDF file containing original western blots for *Figure 1C*, indicating the relevant bands.

**Source data 2.** Original files for western blot analysis displayed in *Figure 1C*.

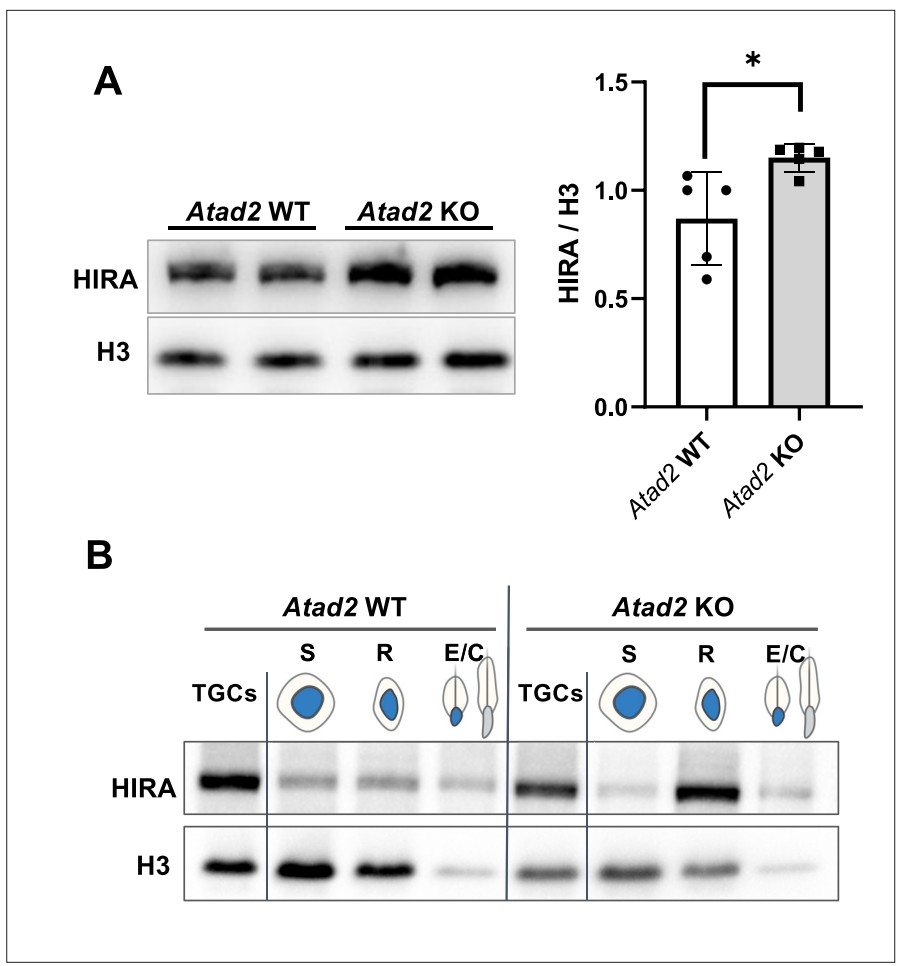

**Figure 2.** *Atad2* gene inactivation leads to the accumulation of HIRA. (**A**) Total protein extracts from the testes of five wild-type (WT) and five *Atad2* KO mice were used to detect HIRA and H3 by immunoblotting. The panel shows a representative immunoblotting after HIRA detection. The histogram in the right panel shows the quantification of HIRA immunoblotting signal normalized to H3. The mean ± standard deviation was 0.86±0.21 for wild type and 1.15±0.06 for *Atad2* KO samples. The *P* value was 0.02 for unpaired Student *t*-test. (**B**) Protein extracts from fractionated spermatogenic cells from wild-type (*Atad2* WT) and knockout (*Atad2* KO) mice (pool of testes from three individuals per genotype) were used to detect HIRA and H3 by immunoblotting.

The online version of this article includes the following source data for figure 2:

**Source data 1.** PDF file containing original western blots for *Figure 2A*, indicating the relevant bands.

**Source data 2.** Original files for western blot analysis displayed in *Figure 2A*.

**Source data 3.** PDF file containing original western blots for *Figure 2B*, indicating the relevant bands.

**Source data 4.** Original files for western blot analysis displayed in *Figure 2B*.

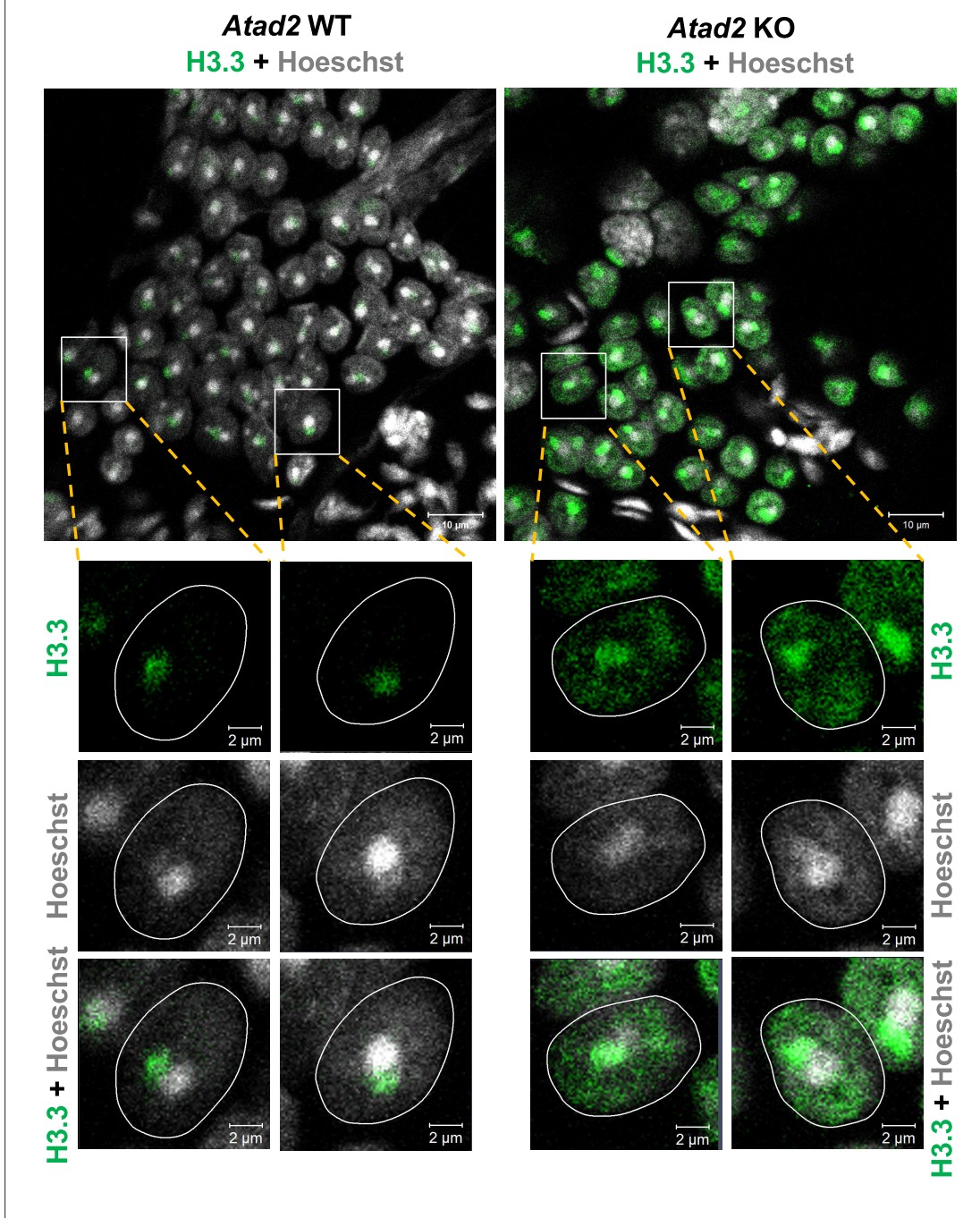

**Figure 3.** Enhanced H3.3 detection in *Atad2* KO post-meiotic cells. A specific antibody against H3.3 was used to detect H3.3 in round spermatid cells from wild-type (*Atad2* WT) and knockout (*Atad2* KO) mice. Round spermatids are recognizable by their distinctive Hoechst-bright chromocenter (upper panels). The lower panels represent a higher magnification of selected cells that are displayed at lower magnification in the upper panels. The images show H3.3 labeling and Hoechst staining alone or the merged images of these staining as indicated. The nuclear boundaries defined by Hoechst staining are outlined by white lines.

Interestingly, visualization of H3.3, the major target of HIRA, by immunofluorescence, in *Atad2* KO round spermatids, displayed a diffuse staining with local concentration (*Figure 3*). H3.3 is normally concentrated on the sex chromosomes (*van der Heijden et al., 2007*; *Fontaine et al., 2022*) in the wild-type cells. Indeed, as expected and previously reported, in the wild-type round spermatids, H3.3 was mostly observed in a single domain, close to the unique chromocenter in these cells (*Figure 3*,

left panels). In the absence of ATAD2, in agreement with an increased residence time of HIRA on chromatin in general and a reduced H3.3 turnover, an increased level of H3.3 could be observed on the sex chromosome (adjacent to the chromocenter), and more interestingly, on all over the genome (*Figure 3*, right panels).

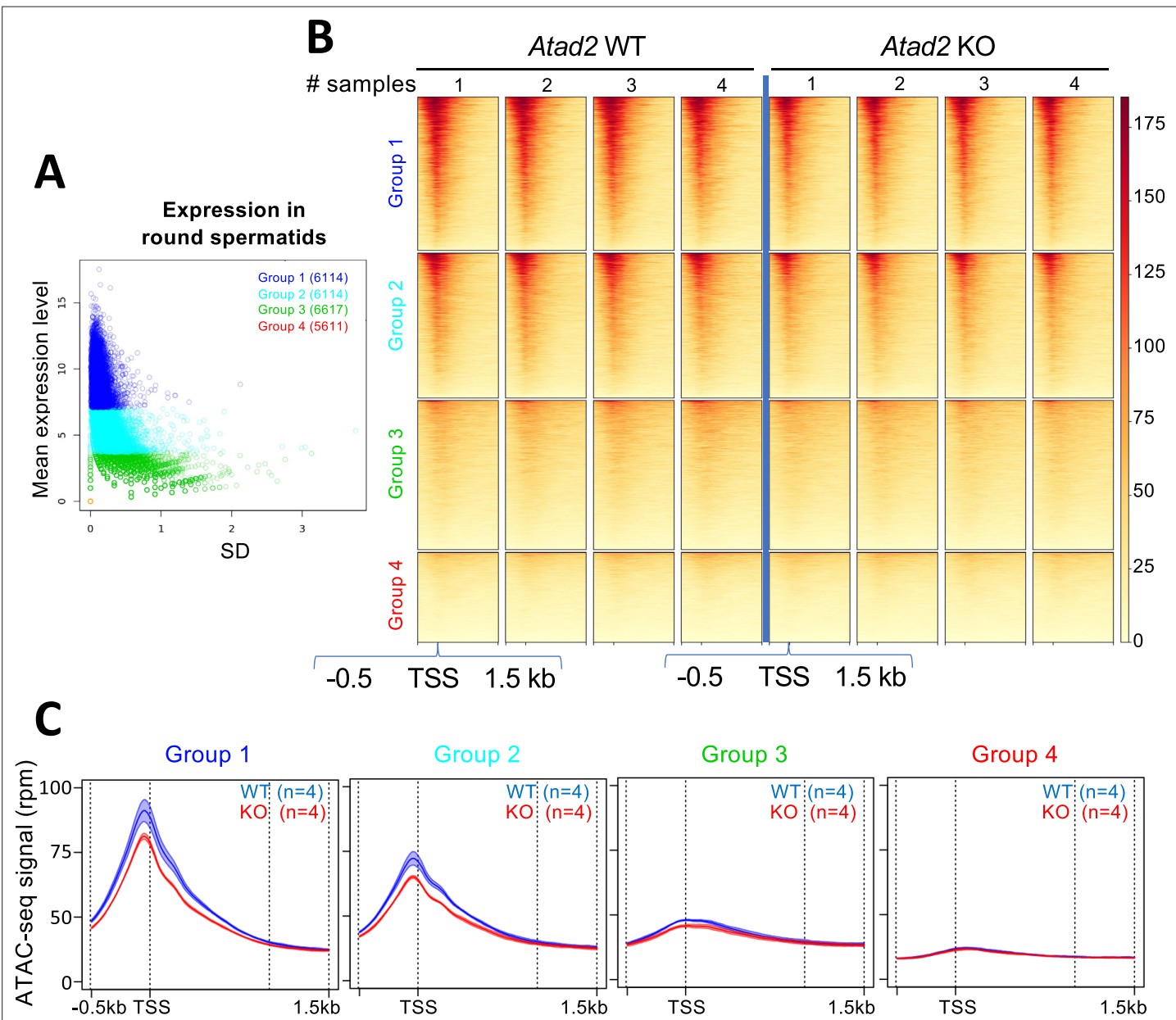

**Figure 4.** ATAD2 controls active gene TSS accessibility. (**A**) Gene selection according to expression in wild-type round spermatids. The plot displays the mean expression levels (y-axis, labeled as "mean expression level") versus the corresponding standard deviations (x-axis), both calculated from three independent biological replicates of isolated round spermatids. The standard deviation reflects the variability of gene expression across biological replicates. Genes were grouped into four categories (Group 1: blue, Group 2: cyan, Group 3: green, Group 4: orange) according to the quartile of their mean expression. For Group 4, all genes have no detectable expression, resulting in a mean expression of zero and a standard deviation of zero; consequently, the 5611 genes in this group are represented by a single overlapping point (red open circle) at the origin. (**B**) Heatmap on ATAC-seq signal. The heatmap shows the ATAC-seq signal of each group (Group 1 in blue, Group 2 in cyan, Group 3 in green, and Group 4 in orange) in rows, for 4 *Atad2* WT and 4 *Atad2* KO samples in columns. (**C**) Profiles according to selected genes. The mean profiles are displayed together with their variability (±2 s.e.m.) across the four replicates for both *Atad2* WT (blue) and *Atad2* KO (red). For groups 1, 2, and 3, the envelopes of the curves remain clearly separated around the peak, indicating a consistent difference in signal between the two conditions. In contrast, group 4 does not present a strong signal and, accordingly, no marked difference is observed between *Atad2* WT and KO in this group.

## General chromatin accessibility and transcriptional regulation by ATAD2

H3.3 is known to be dynamically incorporated on a group of active genes, especially around their transcriptional start sites (TSS) (*de Dieuleveult et al., 2016*). We therefore predicted that the observed increased H3.3 residence time in the absence of ATAD2 would decrease TSS regions' accessibility, especially those of highly active genes. To verify this prediction, we purified round spermatid fractions from wild-type and *Atad2 KO* mice testes and measured the general chromatin accessibility following an ATAC-seq approach. Chromatin accessibility was then assessed around the TSS of genes, which were categorized into four groups based on the quartile of their mean expression from three independent transcriptomes of wild-type round spermatids (*Figure 4A*), from the top 25% most highly expressed genes (group 1) to the 25% least expressed genes (group 4). As predicted, the absence of ATAD2 was associated with a significant decrease in TSS region accessibility of the most active genes (*Figure 4B and C*), supporting a role for ATAD2 in maintaining chromatin dynamics at the most highly active chromatin regions.

To visualize the genes more sensitive to ATAD2-mediated chromatin remodeling that affect transcription, we monitored whole genome transcriptional activity during the first spermatogenic wave at day 20 post-partum (PP), when meiosis is complete, and in the first post-meiotic cells that appear on 22, 24, and 26 days PP.

In order to definitively confirm our working hypothesis on the ATAD2-HIRA-H3.3 functional interaction, we took advantage of published H3.3-dependent transcriptomic data (*Fontaine et al., 2022*). Indeed, the authors of this publication identified a series of genes that are misregulated in cells with inactivated *H3.3b* gene. H3.3 is encoded by two distinct genes, *H3f3a* and *H3f3b*, which are located on different chromosomes and display distinct genomic organization, yet produce proteins with identical amino acid sequences (*Krimer et al., 1993*). Because of the post-meiotic major activation of *Atad2*, we specifically focused on the gene lists that show a H3.3-dependent deregulated expression in post-meiotic round spermatids.

In these transcriptomes, we specifically followed the expression of the published lists of genes that are downregulated or upregulated in the absence of H3.3 (*Fontaine et al., 2022*). Very interestingly, we observed that a significant number of genes (37%) that were reported to be down-regulated in the absence of H3.3 (H3.3-activated genes) are upregulated in the absence of ATAD2, particularly at day 26 PP, when post-meiotic cells are present in the testes (*Figure 5A*, *Figure 5—figure supplement 1*). Gene Set Enrichment Analysis (GSEA) applied to genes that are differentially expressed between wild-type and *Atad2* KO cells at day 26 PP confirmed that the 'H3.3-activated' gene set (corresponding to the published list of H3.3-activated genes) is clearly enriched in *Atad2* KO cells (*Figure 5A*, right panel).

This regulation can also be observed when we considered the published list of genes that were up-regulated in the absence of H3.3 (H3.3-repressed genes). In this case, the absence of ATAD2 led to the enhancement of the down-regulation of these genes (37%). In other words, the absence of ATAD2 led to an enhancement of H3.3-dependent gene repression (*Figure 5B*, *Figure 5—figure supplement 1*).

Fontaine and colleagues also reported that sex chromosome-linked genes that normally undergo Meiotic Sex Chromosome Inactivation (MSCI) were up-regulated in the absence of H3.3, indicating that H3.3 is required for their repression. We also investigated these genes and showed that they present an enhancement of repression in the absence of ATAD2 (51%) (*Figure 5C*, *Figure 5—figure supplement 1*).

These analyses clearly indicate that the absence of ATAD2 enhances H3.3 functions in agreement with increased residence time of H3.3 in nucleosomes. Genes that require H3.3 to be repressed or activated present an enhanced repression or activation, respectively, in the absence of ATAD2.

Overall, these investigations remarkably confirmed a role for ATAD2 in regulating H3.3 function.

## ATAD2 is required for controlled histone eviction and PRM assembly

ATAD2's involvement in HIRA-chromatin interaction mobility and its role in H3.3 assembly in post-meiotic cells, along with its similar function in the control of other histone chaperones like FACT (*Wang et al., 2021*), led us to investigate whether ATAD2 participates in controlling the histone-to-protamine replacement that occurs later in these cells. The histone eviction process involves a genome-wide histone H4 hyperacetylation (*Shiota et al., 2018*) and a TP-mediated PRMs assembly,

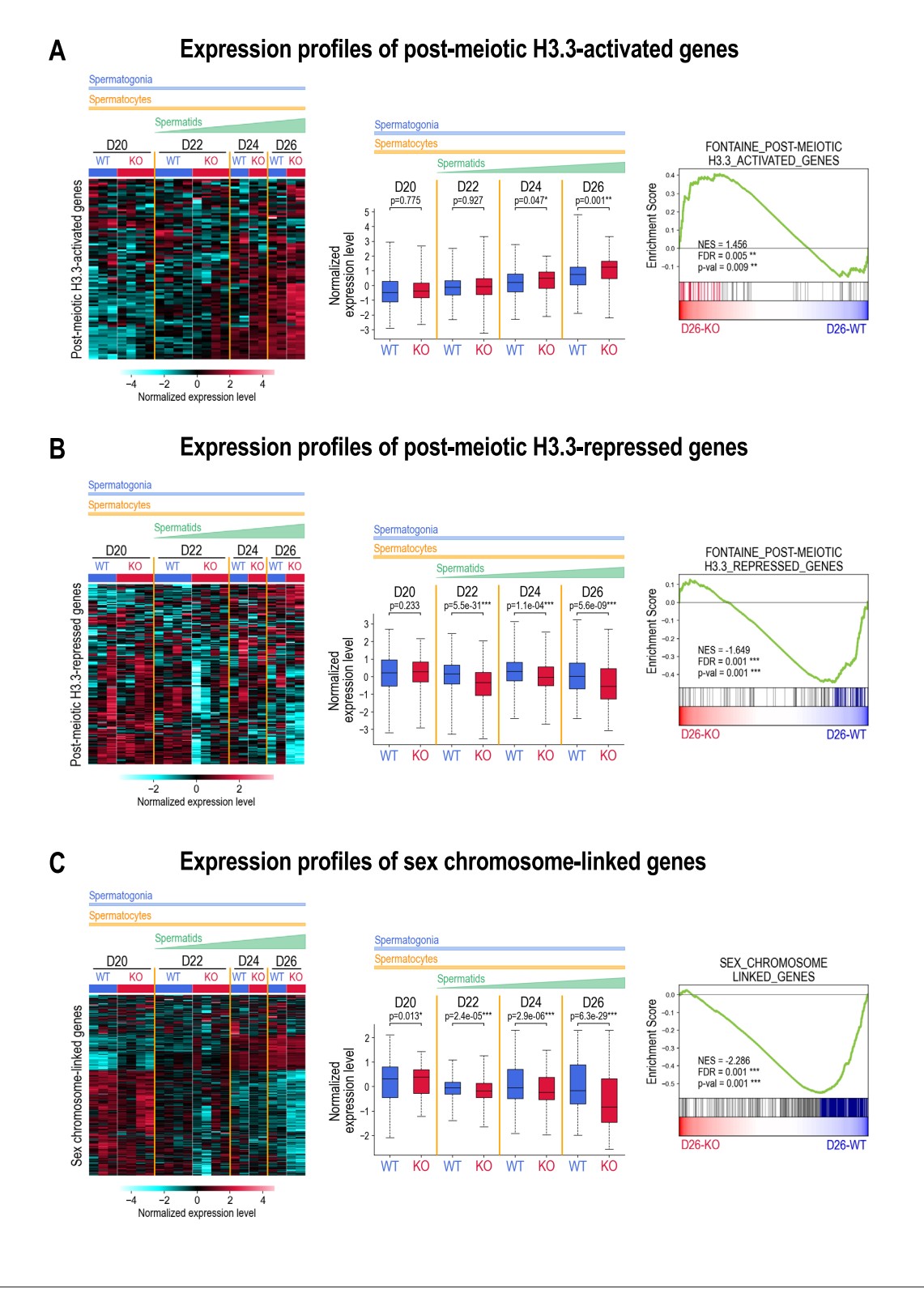

**Figure 5.** ATAD2 enhances H3.3 gene regulatory functions. (**A**) Expression profiles of post-meiotic H3.3-activated genes. The heatmap (left panel) displays the normalized expression levels of genes identified by Fontaine and colleagues as upregulated in the absence of histone H3.3 (***Fontaine et al., 2022***) for *Atad2* WT (WT) and *Atad2* KO (KO) samples at days 20, 22, 24, and 26 PP (D20 to D26). The color scale represents the z-score of log-transformed DESeq2-normalized counts. The middle panel box plots display pooled, normalized expression levels, aggregated across replicates and

*Figure 5 continued on next page*

*Figure 5 continued*

genes, for each condition (WT and KO) and each time point (D20 to D26). Statistical significance between WT and KO conditions was determined using a two-sided *t*-test, with p-values indicated as follows: * for *P*-value <0.05, ** for *P*-value <0.01 and *** for *P*-value <0.001. The right panel shows the results of gene set enrichment analysis (GSEA), which assesses whether predefined groups of genes show statistically significant differences between conditions. Here, the post-meiotic H3.3-activated genes set, identified by *Fontaine et al., 2022*, is significantly enriched in *Atad2* KO compared with WT samples at day 26 (*P*<0.05, FDR <0.25). Colored vertical bars indicate the "leading edge" genes (i.e., those contributing most to the enrichment signal), located before the point of maximum enrichment score. (**B**) As shown in (**A**) but for the "post-meiotic H3.3-repressed genes" gene set. (**C**) As shown in (**A**) but for the " sex chromosome-linked genes " gene set.

The online version of this article includes the following figure supplement(s) for figure 5:

**Figure supplement 1.** Venn diagrams showing the overlap of significantly differentially expressed genes between *Atad2* KO and *Atad2* WT samples at day 26 post-partum (PP), compared with three gene sets reported by *Fontaine et al., 2022*.

which directs histone eviction and the final packaging of the male genome (*Barral et al., 2017a*). To test the involvement of ATAD2 in spermatogenic cell chromatin reorganization, we sought to co-detect testis-specific H2B variant, TH2B (*Montellier et al., 2013*), along with transition protein 1 (TP1) and protamine 1 (PRM1). TH2B is first expressed at the commitment of spermatogenic cells into meiotic divisions and progressively becomes the major H2B species of spermatogenic cell chromatin in the subsequent stages (*Montellier et al., 2013*). As expected, the accumulation of TP1 and PRM1 in wild-type cells is associated with the eviction of the majority of histones, reflected by the mutually exclusive labeling of TH2B and TP1/PRM1 in distinct cells (*Figure 6A and B*, *Atad2* wild-type mice). In contrast, in *Atad2* KO cells, TH2B and TP1/PRM1 were co-detected in a significant number of cells (*Figure 6A and B*), suggesting that the process of histone eviction is disturbed. However, in *Atad2* KO cells, the initial delay in histone removal is transient, as spermatids with high levels of transition protein TP1 and PRM1 eventually appear (*Figure 6*).

These observations strongly suggest that, in the absence of ATAD2, the process of histone eviction is also slowed down or delayed, allowing for the simultaneous detection of histones and histone-replacing proteins in the same cells.

Given that NUT-CBP/p300-dependent H4 hyperacetylation, along with the expression of H2A.L.2, TPs, and PRMs, is essential for histone removal, we also investigated whether the observed delay in histone removal is due to defective histone acetylation, particularly at H4K5, as well as disrupted cell type-specific expression and localization of H2A.L.2, TPs, and PRMs. Immunohistochemical detection of H4K5ac, H2A.L.2, TP2, PRM1, and PRM2 revealed a similar pattern and cell type-specific expression of the investigated proteins when comparing *Atad2* KO and wild-type post-meiotic cells. (*Figure 6—figure supplement 1*).

These data support the idea that the perturbed modified flexibility of histone/TP/PRM chaperones does not grossly affect the other molecular events involved in the histone-to-PRM replacement.

## Perturbed pre-PRM2 processing in the absence of ATAD2

We have shown previously that the H2A.L.2-dependent process of incorporation of TPs into nucleosomes is required for the proper processing of pre-PRM2. The abnormal overlapping staining of histones, TP, and protamines observed by immunofluorescence in *Atad2* KO cells suggested that TPs may not effectively mediate the pre-PRM processing. To test this hypothesis, we took advantage of our specific pre-PRM2 antibody (*Rezaei-Gazik et al., 2022*) to investigate pre-PRM2 maturation and assembly. *Figure 7A* shows that although pre-PRM2 is present in both wild-type and *Atad2* KO elongating spermatids, in the absence of ATAD2, pre-PRM2 tends to form dense aggregates as well as fragmented genomic domains. This phenomenon is also associated with an increased accumulation of pre-PRM2 protein level in testis extracts of *Atad2* KO spermatogenic cells compared to wild-type cells as revealed by ELISA and immunoblotting (*Figure 7B*, *Figure 7—figure supplement 1*).

These data provide strong evidence that ATAD2 is required to streamline the activity of histone chaperones as well as other factors involved in histone exchange and TP/PRM assembly.

## Defective mature spermatozoa genome organization and male fertility in vitro

We show that although the *Atad2* KO post-meiotic cells' defective chromatin organization does not affect the testis histology (*Figure 6—figure supplement 1* and **data not shown**), it impairs testis

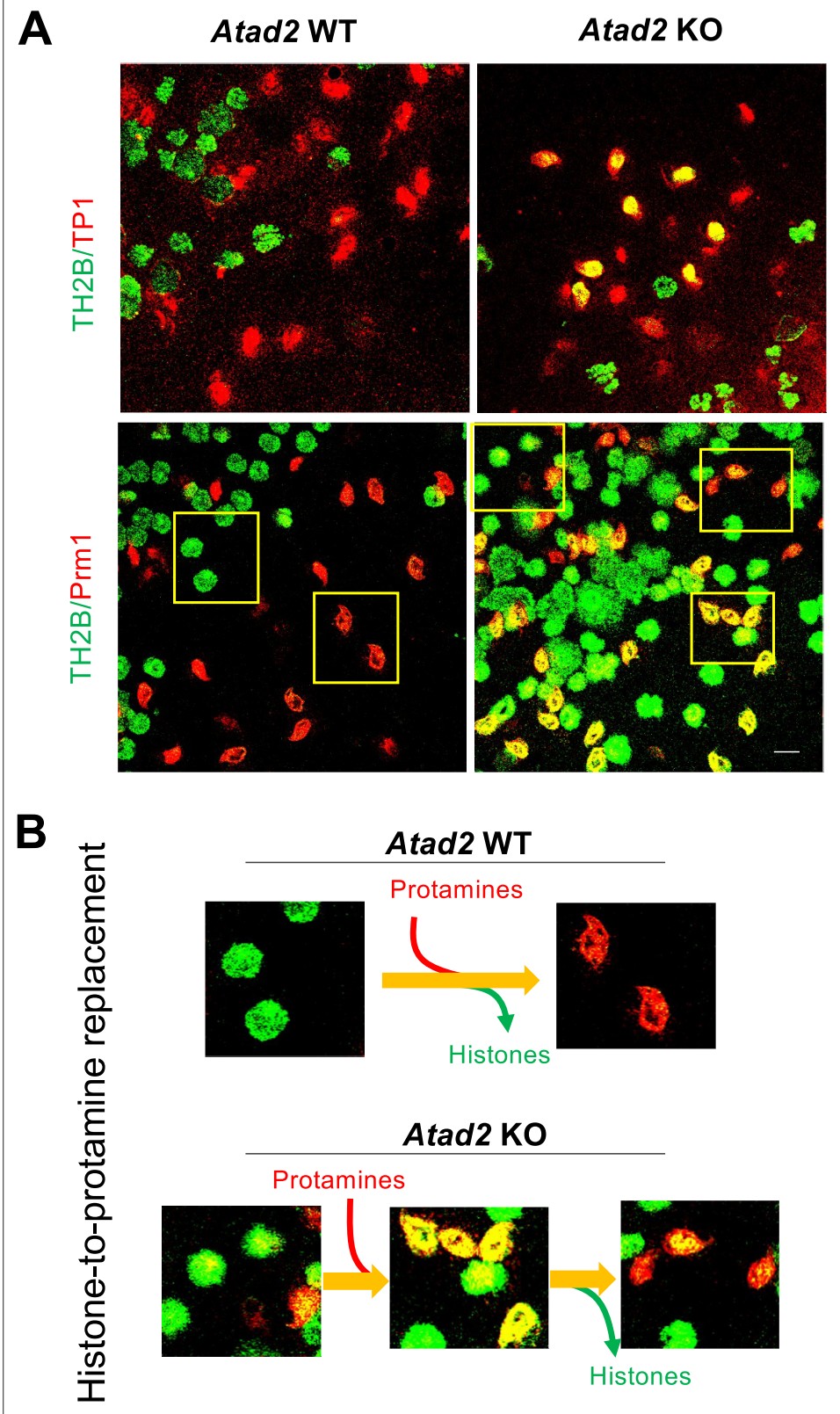

**Figure 6.** ATAD2 is required for efficient histone-to-protamine replacement. (**A**) Spermatogenic cell preparations from wild-type and *Atad2* KO mice were stained with antibodies against TH2B (in green) and transition protein 1 (TP1) or protamine 1 (PRM1) (in red). A yellow signal indicates the co-detection of TH2B along with TP1 or PRM1. The indicated fields are shown at higher magnification in panel "B". Scale bar: 10 µm. (**B**) Fields shown in panel

*Figure 6 continued on next page*

*Figure 6 continued*

"A" are ordered to illustrate the transition from the histone-associated genome to its packaging by protamines, considering both wild-type and *Atad2* KO spermatogenic cells.

The online version of this article includes the following figure supplement(s) for figure 6:

**Figure supplement 1.** *Atad2* gene inactivation does not disrupt testis morphology, acetylation of H4K5, or the cell-specific accumulation of H2A.L.2, TP2, PRM1, and PRM2.

**Figure supplement 2.** *Atad2* depletion affects testis weight and mature sperm counts.

weight and sperm count, which are significantly reduced in *Atad2* KO mice (*Figure 6—figure supplement 2*).

As expected, the impact of ATAD2 on HIRA function and TP/PRM assembly also has an impact on proper sperm genome compaction. Indeed, mature sperm genome decompaction test based on the incubation of sperm heads with heparin and the reducing agent DTT revealed a significant difference in the ability of the genome to resist genome decompaction conditions between wild-type *Atad2* and *Atad2* KO spermatozoa (*Figure 8A*; *Figure 8—figure supplement 1*). Given that spermatozoa with defective genome compaction are known to have poor success rate in in vitro fertilization (*Bashiri et al., 2021*), we expected that spermatozoa from *Atad2* KO mice would perform worse in in vitro fertilization assays compared to wild-type spermatozoa. *Figure 8B* shows that this is, in fact, the case. However, surprisingly, fertility tests carried out on sexually mature adult males revealed that *Atad2* KO male mice are fertile, with only a statistically non-significant effect observed on their ability to foster pups (*Figure 8C*), indicating that in vivo fertilization could buffer a relatively large series of sperm genome compaction defaults.

## Discussion

Taking advantage of the very conserved structural features of ATAD2 and relying on the power of yeast genetics, we previously discovered that ATAD2's function is mostly to control major histone chaperone functions. Using a screen of the Abo1/ATAD2-dependent arrest of *S. pombe* cell growth, we found that impressively, 11 of the 14-growth arrest suppressor strains that we isolated impacted histone chaperones or histones themselves, with 8 distinct inactivating mutations in the histone chaperone HIRA complex. Furthermore, investigation of ATAD2-depleted mouse ES cells showed that HIRA accumulates and remains bound to nucleosomes, leading to nucleosome overloading around active gene TSSs and on the normally nucleosome-free regions (NFRs) of these genes.

Here, we showed that *Atad2* is highly expressed in post-meiotic spermatogenic cells and that in its absence, as in ES cells and in different cancer cells, HIRA accumulates. Our new data suggest that in the absence of ATAD2 activity, HIRA residency time on chromatin increases, leading to decreased turnover and overloading of its substrate, the histone variant H3.3 (*Figure 9*). This replication-independent histone has been first shown to assemble on the transcriptionally inactive meiotic cell sex chromosomes by Peter de Boer's laboratory (*van der Heijden et al., 2007*). Subsequent studies showed that H3.3 remained enriched on the sex chromosomes also in early post-meiotic cells, usually localized close to the unique large chromocenter in these cells (*Fontaine et al., 2022*).

In agreement with the proposed role for ATAD2 in the control of HIRA-dependent dynamics of H3.3, we observed a more intense labeling of H3.3 on the sex bodies in round spermatids and, impressively, an increase in the genomic background detection of H3.3 in these cells.

The role of ATAD2 in the HIRA/H3.3 functions is further supported by comparing the reported H3.3-dependent transcriptome with the ATAD2-dependent transcriptome identified here.

Indeed, in the absence of ATAD2, we observed an enhancement of H3.3-dependent gene regulatory events (activation or repression), in agreement with an increased H3.3 presence and hence function, in the chromatin of *Atad2* KO post-meiotic cells.

These investigations on ATAD2's role in post-meiotic spermatogenic cells built upon our previous findings in *S. pombe* and ES cells, uncover an additional level of regulation of histone turnover, which relies on the modulation of histone chaperone–chromatin interactions. Previously, A. Groth's laboratory discovered that the histone chaperone DNAJC9 recruits the ATPase HSP70 to provide the necessary energy to release the chaperone from chromatin (*Hammond et al., 2021*). Considering our data on the role of the ATPase ATAD2 with respect to HIRA and that of HSP70 with respect to DNAJC9, it

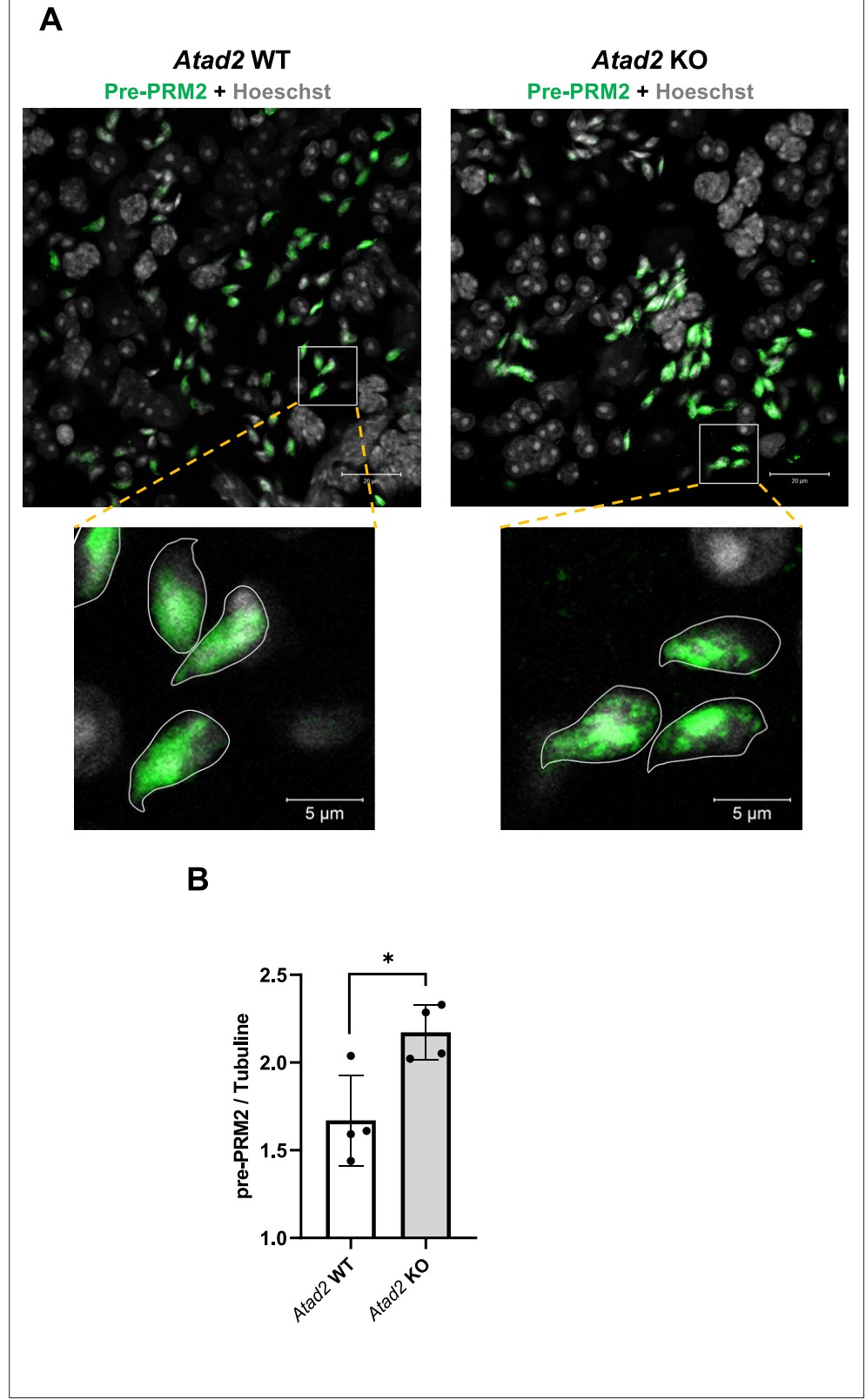

**Figure 7.** ATAD2 is required for proper pre-protamine 2 processing and protamine assembly. (**A**) A specific antibody raised against the processed part of pre-PRM2 (*Rezaei-Gazik et al., 2022*) was used to detect pre-PRM2 expression and localization in tubular elongating spermatid cells. The lower panels represent the magnification of selected cells. (**B**) The amount of pre-PRM2 was detected using soluble extracts from testis of wild-type and

*Figure 7 continued on next page*

*Figure 7 continued*

*Atad2* KO mice using ELISA. Histograms show the normalized values (pre-PRM2/tubulin) of 4 measurements of independent sedimentation samples of Elongating/Condensing fractions. Mean ± Standard Deviation is 1.6±0.2 and 2.1±0.1 for wild type and *Atad2* KO samples, respectively. The p-value obtained using an unpaired Student's *t*-test was 0.0158.

The online version of this article includes the following source data and figure supplement(s) for figure 7:

**Figure supplement 1.** ATAD2 is required for proper pre-protamine 2 processing.

**Figure supplement 1—source data 1.** PDF file containing original western blots for *Figure 7—figure supplement 1*, indicating the relevant bands.

**Figure supplement 1—source data 2.** Original files for western blot analysis displayed in *Figure 7—figure supplement 1*.

---

is tempting to propose that ATP energy is required for the proper function of histone chaperones in controlling histone turnover.

Interestingly, very recently, the Song laboratory demonstrated that the *S. pombe* Abo1/ATAD2 is required for ATP-dependent dissociation of the histone chaperone FACT from chromatin, particularly at TSSs (*Jang et al., 2024*). In line with this conclusion, ChIP-seq mapping of FACT in ATAD2-depleted ES cells revealed an accumulation of nucleosome-bound FACT at the TSSs of active genes (*Wang et al., 2021*), supporting the idea that ATAD2 not only regulates the dynamics of histone-bound HIRA but also that of other histone chaperones such as FACT.

In addition to histones, many chromatin-bound regulatory factors also need to dynamically interact with chromatin, and hence ATAD2 could also be involved in mediating the dynamic interactions of such chromatin-bound factors. In agreement with this hypothesis, using proteomics of ATAD2-bound nucleosomes in ES cells, we previously co-purified various chromatin-bound factors (*Morozumi et al., 2016*). Other AAA+ATPase factors, such as p97/VCP, have also been implicated in the remodeling of chromatin-bound factors. Indeed, the AAA+ATPase factor, p97/VCP, has been shown to remove H3 variant CENP-A (*van den Berg et al., 2023*), the chromatin trapped PARP1 (*Krastev et al., 2022*), trapped Ku70/80 (*van den Boom et al., 2016*), DNA damage sensors, DDB2 and XPC (*Puumalainen et al., 2014*), RNA pol II (*Lafon et al., 2015*), and other chromatin interacting factors. Similarly, ATAD2 AAA+ATPase very likely acts on other chromatin-bound factors than histone chaperones. However, in contrast to p97/VCP, and due to its bromodomain, ATAD2 would preferentially act on H4K5 acetylated chromatin regions and would specifically ensure the dynamics of H4 acetylated chromatin-bound factors. Based on the data presented here, we can also include a role for ATAD2 to streamline steps involving the H2A.L.2-dependent recruitment of TPs and protamine assembly and the final histone displacement in spermatids. The chaperones responsible for TP/PRM assembly are not yet identified, but it is possible that ATAD2 plays a role in dissociating these chaperones from their substrate proteins after the assembly process. That is why in our *Atad2* KO spermatids, we observed an overlap between TP/PRM and histones. This could be due to the delayed removal of chaperones bound to TPs and PRMs and the inability of pre-PRM2 to be processed and hence of PRMs to efficiently replace histones. Since histone replacement still occurs in the absence of ATAD2, it suggests that other AAA+ATPases, such as p97/VCP, may compensate for its function. However, under these conditions, although most of the histone replacement takes place, our functional tests such as in vitro chromatin decompaction and in vitro fertilization strongly suggest that mature spermatozoa genome organization is not fully complete and the spermatozoa nuclei remain fragile and sensitive to external assaults.

In summary, the functional analysis of ATAD2 during its physiological expression in the post-meiotic phases of spermatogenesis supports our previous conclusions regarding its role in ES cells and its conserved activity in *S. pombe*. Together, these studies reveal a novel layer of regulation in histone turnover and chromatin dynamics, based on the control of histone-bound histone chaperone release.

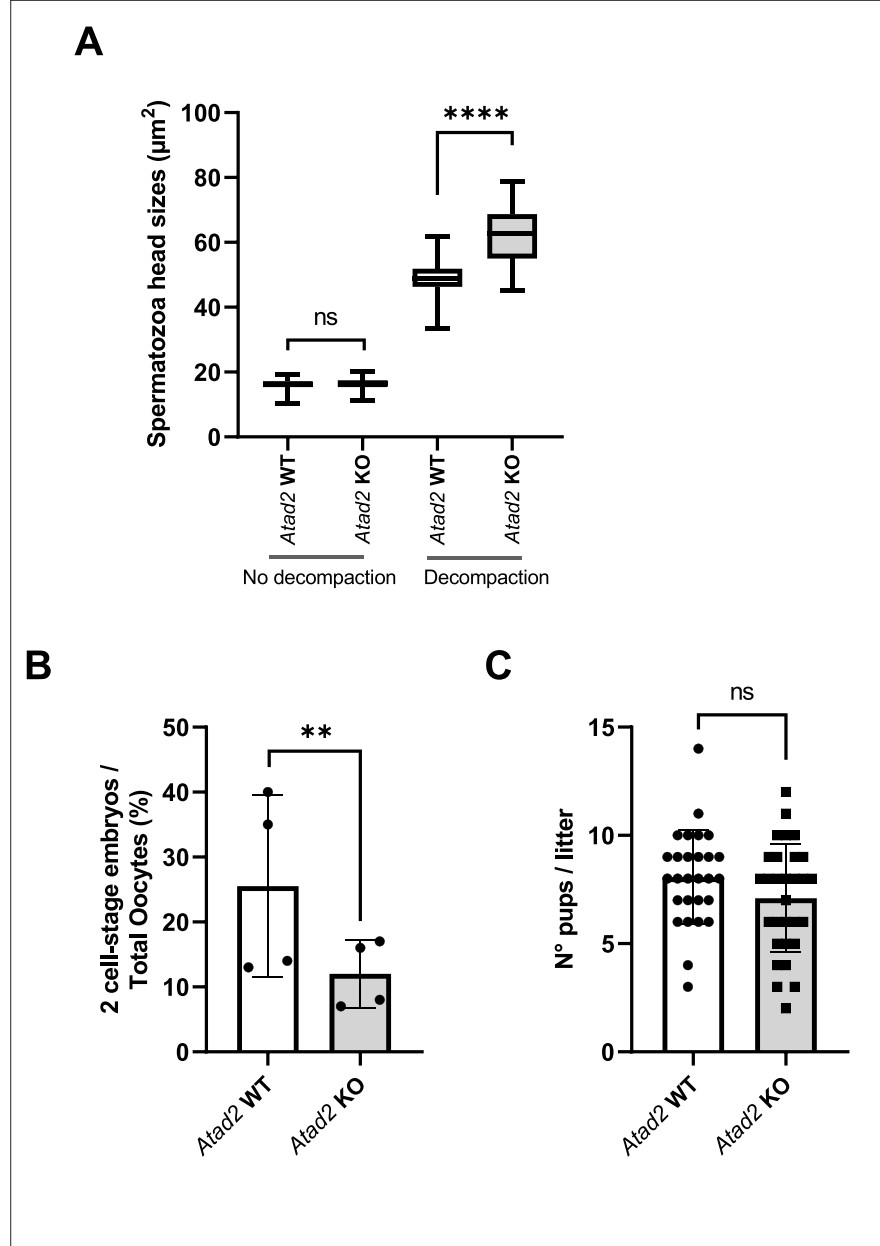

**Figure 8.** ATAD2s' function in sperm genome compaction and fertility parameters. (**A**) Epididymal sperm cells were isolated by the swim-up method from wild-type or *Atad2* KO mice and underwent a decompaction test and the sizes of the sperm heads were measured and represented as box plots (n=75 sperm heads / box plot). The value ranges (mean) were 10.4~19.3 (16.0) for wild type sperms without decompaction; 11.4~20.1 (16.4) for *Atad2* KO sperms without decompaction; 33.4~61.8 (48.5) for wild type sperms after decompaction; 45.0~78.9 (62.6) *Atad2* KO sperms after decompaction. Tukey's HSD post two-way ANOVA test was used. ****$P$<0.001. (**B**) Pools of oocytes from C57BL6 females were obtained and used for in vitro fertilization (IVF) with spermatozoa from WT or *Atad2* KO males (the experiment was repeated 4 times independently). The fertilization success rate is indicated as a percentage of oocytes giving rise to stage 2 embryos. Mean ± Standard Deviation is 25.5±14 and 12±5.2 for wild type and *Atad2* KO experiments, respectively, and the $P$=0.002 for ratio paired Student *t*-test. (**C**) Five wild-type and five *Atad2* KO male mice were each mated with two C57BL6 females, the resulting litter sizes were recorded and represented as box plots (n=28 and 30, respectively). The average numbers of pups are respectively 8.0±2.1 and 7.1±2.5 and the *P-value of* Student *t*-test=0.1.

The online version of this article includes the following figure supplement(s) for figure 8:

**Figure supplement 1.** Sperm head decompaction after DTT and heparin treatment in wild-type and Atad2 KO males.

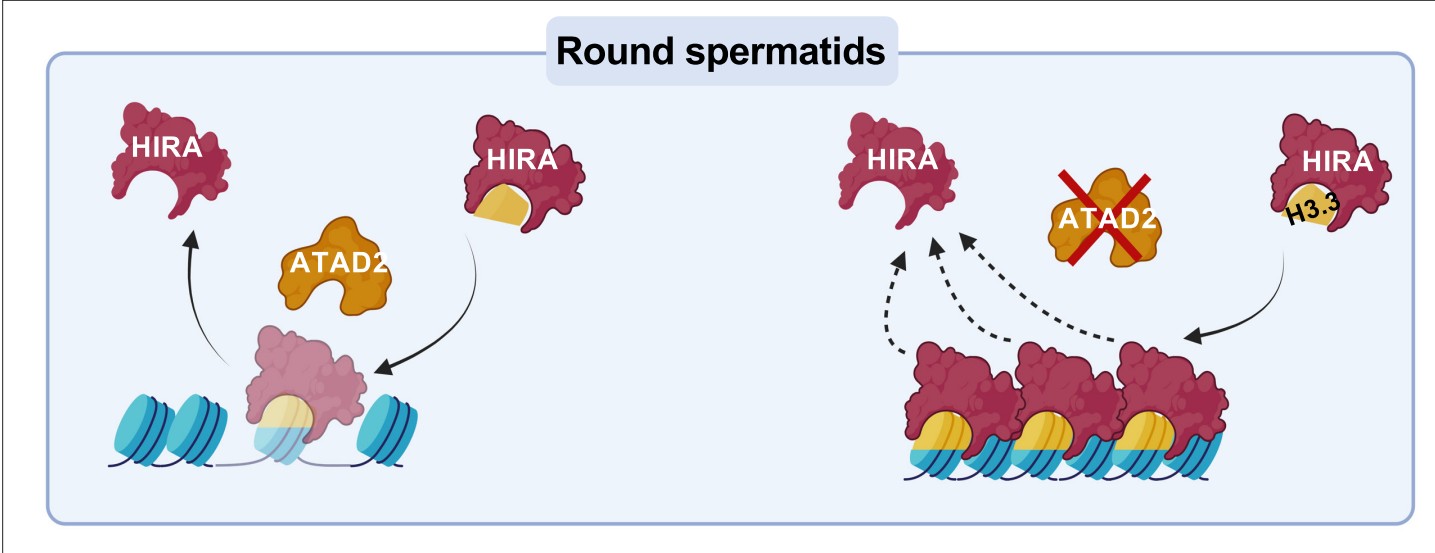

**Figure 9.** A model summarizing the role of ATAD2 in regulating chromatin-bound HIRA dynamics (along with other histone and non-histone protein chaperones) during the post-meiotic stages of mouse spermatogenesis. Following the deposition of H3.3 by HIRA onto chromatin, ATAD2 facilitates the release of HIRA. In the absence of ATAD2, both HIRA and the newly incorporated H3.3 persist longer in nucleosomes. The scheme was created using BioRender (https://BioRender.com/Cmcjckk).

# Materials and methods

## Key resources table

| Reagent type (species) or resource | Designation | Source or reference | Identifiers | Additional information |
|---|---|---|---|---|
| Genetic reagent (*Mus musculus*) | *Atad2* knockout-first' allele | ***Skarnes et al., 2011*** | KO mice | International KO Mouse Consortium; RRID:MGI:4432934 |
| Genetic reagent (*Mus musculus*) | *Atad2*$^{LacZΔNeo-Exon12}$ | This paper | KO mice | *Atad2* ko-first' allele mice were crossed with CMV-Cre mice; RRID:MGI:4432934 |
| Sequence-based reagent | *Atad2*_84231_F | Eurogentec | Forward primer | TATCCAACAAGCCTGAGCCC |
| Sequence-based reagent | *Atad2*_84231_R | Eurogentec | Reverse primer | CAACTGGAGCTGGGTCTTCC |
| Sequence-based reagent | Cas-R | Eurogentec | Reverse primer | TCGTGGTATCGTTATGCGCC |
| Chemical compound, drug | Bovine Serum Albumin | Euromedex | Cat# 04-100-812-C | |
| Commercial assay or kit | Clarity Western ECL Substrate | Bio-Rad | Cat # 170-5061 | |
| Commercial assay or kit | Clarity Max Western ECL Substrate | Bio-Rad | Cat # 170-5062 | |
| Chemical compound, drug | Complete Protease inhibitor EDTA-free | Sigma-Aldrich | Cat# 4693159001 | |
| Other | Dako Fluorescent Mounting Medium | Dako | Cat# S3023 | |
| Chemical compound, drug | DL-Dithiothreitol | Euromedex | Cat# EU0006 | |
| Chemical compound, drug | DNase I recombinant | Roche | Cat# 4536282001 | |
| Chemical compound, drug | DPBS | Gibco | Cat# 14200-075 | |
| Chemical compound, drug | EDTA | Sigma-Aldrich | Cat# E6635 | |
| Chemical compound, drug | EGTA | Sigma-Aldrich | Cat# E3889 | |
| Chemical compound, drug | Formalin solution, neutral buffered, 10% | Sigma-Aldrich | Cat# HT5012 | |
| Chemical compound, drug | Glutaraldehyde solution | Sigma-Aldrich | Cat# G5882 | |

*Continued on next page*

*Continued*

| Reagent type (species) or resource | Designation | Source or reference | Identifiers | Additional information |
|---|---|---|---|---|
| Chemical compound, drug | Heparin sodium salt | Sigma-Aldrich | Cat# H3149 | |
| Other | Hoechst | Invitrogen | Cat# 33342 | |
| Chemical compound, drug | Human Chorionic Gonadotropin | MED-VET | Cat# Chorulon | 1500 UI |
| Chemical compound, drug | Laemmli Sample Buffer | BioRad | Cat# 1610747 | |
| Chemical compound, drug | Magnesium Chloride | Euromedex | Cat# 2189 | |
| Chemical compound, drug | Mayer's Hematoxylin Solution | Sigma-Aldrich | Cat# MHS16 | |
| Chemical compound, drug | M2 medium | Sigma-Aldrich | Cat# M7167 | |
| Chemical compound, drug | M16 medium | Sigma-Aldrich | Cat# M7292 | |
| Other | Nitrocellulose Blotting membrane | Amersham Protran | Cat# 10600015 | |
| Other | Nonidet P 40 Substitute | Sigma-Aldrich | Cat# 74385 | |
| Other | NuPAGE 4–12% Bis-Tris Gel | Invitrogen | Cat# NP0335BOX | |
| Chemical compound, drug | Potassium Ferricyanide | Sigma-Aldrich | Cat# P-8131 | |
| Chemical compound, drug | Potassium Ferrocyanide | Sigma-Aldrich | Cat# P-9387 | |
| Chemical compound, drug | Pregnant Mare Serum Gonadotropin (PMSG) | MED-VET | Cat# Syncro-part PMSG 600 UI | |
| Chemical compound, drug | Saccharose | Sigma-Aldrich | Cat# S7903 | |
| Chemical compound, drug | Saponin from quillaja bark | Sigma-Aldrich | Cat# S7900 | |
| Chemical compound, drug | Sodium deoxycholate | Sigma-Aldrich | Cat# D6750 | |
| Other | Superfrostslides | Thermo Scientific | Cat# LCSF05 | |
| Commercial assay or kit | TMB ELISA Substrate (High Sensitivity) | Abcam | Cat# AB171523 | |
| Chemical compound, drug | Trans-Blot Turbo 5X Transfer Buffer | BioRad | Cat# 10026938 | |
| Chemical compound, drug | Triton X-100 | Sigma-Aldrich | Cat# T9284 | |
| Chemical compound, drug | TRIzol | Thermo Scientific | Cat# 15596026 | |
| Chemical compound, drug | TWEEN 20 | Sigma-Aldrich | Cat# P7949 | |
| Chemical compound, drug | Urea | Sigma-Aldrich | Cat# U5378 | |
| Chemical compound, drug | X-Gal | Invitrogen | Cat# B1690 | |
| Other | 96 Well Round (U) Bottom Plate | Thermo Scientific | Cat# 163320 | |
| Chemical compound, drug | 450 nm Stop Solution for TMB Substrate | Abcam | Cat# AB171529 | |
| Antibody | Anti-H2B testis specific (rabbit polyclonal) | Abcam | Cat# ab178426 | IF (1:100); RRID:AB_3720329 |
| Antibody | Anti-ATAD2 (rabbit polyclonal) | Homemade | This paper | WB (1:1000) generated by Covalab, France |
| Antibody | Anti-H2A.L1/L2 (rabbit polyclonal) | Homemade | SAB771 | IHC (1:200) generated by Covalab, France |
| Antibody | Anti-Histone H3 (rabbit polyclonal) | Abcam | Cat#: Ab1791 | WB (1:5000); RRID:AB_302613 |
| Antibody | Anti-H3F3B (mouse monoclonal) | Abnova | Cat#: H00003021-M01 | IF (1:200); RRID:AB_425473 |
| Antibody | Anti-H4K5Ac (mouse monoclonal) | PTM Biolabs | Cat#: PTM-163 | IHC (1:200); RRID:AB_3717383 |

*Continued on next page*

*Continued*

| Reagent type (species) or resource | Designation | Source or reference | Identifiers | Additional information |
|---|---|---|---|---|
| Antibody | Anti-Hira (D6O8L) (rabbit monoclonal) | Cell Signaling Technology | Cat#: 12463 | WB (1:1000); RRID:AB_2797927 |
| Antibody | Anti-Pre-PRM2 (rabbit polyclonal) | Homemade | No reference | IF (1:100); generated by Covalab, France |
| Antibody | Anti-PRM1 (mouse monoclonal) | BRIAR Patch Biosciences | Cat#: HUP1N | IHC (1:200); RRID:AB_2651186 |
| Antibody | Anti-PRM2 (mouse monoclonal) | BRIAR Patch Biosciences | Cat#: HUP2B | IHC (1:200); RRID:AB_2687949 |
| Antibody | Anti-TNP2 (B-2, mouse monoclonal) | Santa Cruz Biotechnology | Cat#: sc-393843 | IHC (1:200); RRID:AB_3720327 |
| Antibody | Goat anti-rabbit IgG (H+L)-HRP | Bio-Rad | Cat# 170–6515 | WB (1:5000); RRID:AB_11125142 |
| Antibody | Goat anti-mouse IgG (H+L)-HRP | Bio-Rad | Cat# 170–6516 | WB (1:5000); RRID:AB_11125547 |
| Antibody | Alexa Fluor 568 Goat anti-mouse IgG (H+L) | Invitrogen | Cat# A-11004 | IF (1:500); RRID:AB_2534072 |
| Antibody | Alexa Fluor 488 Goat anti-rabbit IgG (H+L) | Invitrogen | Cat# A-11034 | IF (1:500); RRID:AB_2576217 |

## Animal care and breeding

Mice were housed at the Grenoble High Technology Animal Facility (PHTA). Mice were euthanized following a procedure approved by the official ethics committee of the University Grenoble Alpes (COMETH, C2EA-12). The investigators directly involved in care and breeding of mice had an official animal-handling authorization obtained after 2 weeks of intensive training and a final formal evaluation.

## Generation of *Atad2*-KO mice

*Atad2* KO mice were obtained from the International Knockout Mouse Consortium, the Welcome Trust Sanger Institute, UK. The 'knockout-first' allele (tm1a) contains an IRES: *lacZ* trapping cassette and a floxed human β-*actin* promoter-driven *neo* cassette inserted into the intron 12 of the *Atad2* gene (**Skarnes et al., 2011**). We created *Atad2*$^{ΔNeo-Exon\ 12}$ mice by crossing mice bearing the *tm1a* allele with mice expressing Cre recombinase under the control of a ubiquitous human promoter CMV. This cross results in an *Atad2*$^{LacZΔNeo-Exon12}$ knockout mouse with the first 11 exons of *Atad2* being fused to β-gal, leading to a non-functional ATAD2 protein depleted of its bromodomain and AAA+ATPase domain.

For *Atad2* KO mice genotyping, multiplex PCR was used to detect LacZ cassette producing a 393 bp WT band and a 293 bp mutant band using primers *Atad2*_84231_F, TATCCAACAAGCCTGA GCCC, *Atad2*_84231_R, CAACTGGAGCTGGGTCTTCC and Cas-R, TCGTGGTATCGTTATGCGCC.

## Fertility test

Fertility tests were carried out with sexually mature 3-month-old males. Five WT males and five *Atad2* KO males were crossed with two C57BL6 females each, during 3 months, and the litter sizes were monitored during this period.

## In vitro fertilization

Eggs were obtained from 6- to 7-week-old superovulated C57/BL6J females, stimulated by a first injection of 7.5 UI of PMSG (Pregnant Mare Serum Gonadotropin), followed 48 hours later by a second injection of 7.5 UI of HCG (Human Chorionic Gonadotropin). Fourteen hours later, females were euthanized by cervical dislocation, and cumulus-oocyte complexes (COCs) released from the ampulla were collected in 500 µL M2 medium. Spermatozoa from wild type and *Atad2* KO males were harvested by dilaceration of the cauda epididymitis and allowed to swim in 1 mL M2 medium for 10 minutes at 37°C. Two hundred microliters of sperm were transferred into 800 µL M16/2% BSA and capacitated for 80 minutes at 37°C /5% $CO_2$. Finally, 300,000 sperm were simultaneously added to the COCs and incubated in M16 medium at 37°C/5% $CO_2$. After 4 hours of incubation, unbound spermatozoa were

removed by three successive washes with 500 µL of M16. Twenty-four hours after fertilization, unfertilized eggs and two-cell embryos (as an indication of successful fertilization) were scored.

## Protein extraction and immunoblotting

Proteins were extracted in 8 M urea, sonicated at 150 J with a probe sonicator, mixed with 4x Laemli loading buffer and heated for 5 minutes at 95°C. Protein samples were loaded on 4–12% Bis-Tris gels and migrated at 150 V for 1 hour 30 minutes. The transfer step was realized using a nitrocellulose membrane with the Bio-Rad Transfer Turbo machine. Nitrocellulose membrane was blocked in 5% milk PBS-Tween 0.1% solution, washed 3 times for 10 minutes with PBS-Tween 0.1% solution, and incubated overnight at 4°C with specific antibodies listed in **Key resources table**. Membranes were washed three times for 10 minutes with PBS-Tween 0.1% solution and were incubated with HRP secondary antibodies at 1/10,000 dilution for 1 hour at RT under agitation. Revelation was performed using ECL substrates and visualization was performed with the Vilber Fusion FX Chemiluminescence imaging system (Vilber).

## ELISA

Urea protein extracts were diluted in PBS and used to coat 96-well round-bottom plates, followed by overnight incubation at 4°C. Coating solution was removed and wells were washed three times with washing buffer (BSA 1%, PBS-T 0.1%). Wells were blocked with blocking buffer (5% BSA in PBS-T 0.1%) for 2 hours at room temperature under agitation. The blocking buffer was removed and wells were washed 2 times with washing buffer. Primary antibodies for pre-PRM2 (1:1000) and tubulin (1:5000) were diluted into blocking buffer, pipetted into their corresponding wells, and incubated overnight at 4°C under mild agitation. Wells were washed four times with washing buffer, and subsequent incubation with secondary HRP antibodies, diluted 1:5000 in blocking buffer, was performed for 1 hour at room temperature under mild agitation. Wells were washed four times with washing buffer, and TMB ELISA Substrate was added to wells and incubated until the color developed, at which time Stop Solution was added to the wells to stop the reaction. The signal was quantified using a plate reader at 450 nm.

## X-Gal staining

Male mice were euthanized by cervical dislocation. Testes were dissected out, drilled in three points, and incubated overnight at 4°C under agitation in fixative solution (0.5% glutaraldehyde, 2 mM $MgCl_2$ and 5 mM EGTA in PBS). After incubation until equilibration in an equilibration buffer (30% sucrose, 2 mM $MgCl_2$ and 5 mM EGTA in PBS), testes were frozen individually in OCT on dry ice. Frozen blocks were sectioned at 10 µm slices on Superfrost slides and let dry overnight. Slides were fixed with fixation buffer (0.2% glutaraldehyde, 100 mM $MgCl_2$ and 5 mM EGTA in PBS) for 10 minutes, washed with washing buffer (PBS, 0.02% NP-40, 0.01% sodium deoxycholate, and 2 mM $MgCl_2$) for 5 minutes and incubated three times in 50% ethanol for 5 minutes each. Then the slides were stained with staining solution (2 mM $MgCl_2$, 0.02% NP-40, 0.01% Na Doc, 5 mM potassium ferricyanide, 10 mM potassium ferrocyanide, and 0.5 mg/mL X-gal) in a wet chamber for several hours at 37°C. The slides were then washed with washing buffer for 5 minutes and with distilled water, and a counterstaining was performed with hematoxylin. Finally, the slides were washed with running tap water for several minutes and mounted with Dako fluorescent mounting medium.

## Immunofluorescence staining of germ cells

Staged seminiferous tubules were prepared as detailed in *Gaucher et al., 2012*. In order to permeabilize the cells, the slides were placed in 0.5% saponin, 0.2% Triton X-100, and 1×PBS at RT for 15 minutes and washed for 5 minutes with PBS. The preparations were then incubated in 5% milk, 0.2% PBS-Tween blocking buffer, at RT for 30 minutes under agitation. The primary antibodies were diluted in 1% dry milk, 0.2% PBS-Tween buffer. The slides were incubated overnight with this diluted solution of the primary antibodies in a humidified chamber at 4°C and then washed three times for 5 minutes each in the antibodies' dilution buffer. The secondary antibodies were diluted at 1:500 in the same buffer and incubated in a humidified chamber for 30 minutes at 37°C, and then washed as for the primary antibodies. The DNA was counterstained with Hoechst, and the slides were mounted using Dako fluorescent mounting medium.

## Sperm head decompaction test

Spermatozoa from wild type and *Atad2* KO males were harvested by dilaceration of the cauda epididymis and allowed to swim in 1 mL M2 medium for 10 minutes at 37°C. The sperm-containing suspension was transferred into a 1.5 mL Eppendorf tube and centrifuged at 3000 rpm for 8 minutes in 4°C. The sperm pellets were resuspended in PBS and dropped onto glass slides to spread and dry at RT. The decompaction mix (50 mM DTT, 400 IU/mL heparin, and 0.2% Triton in PBS) was added to each dried sperm drop for 2 minutes and the decompaction was stopped by placing the slides in 4% PFA for 15 minutes. Slides were washed with PBS for 5 minutes, stained with DAPI, and mounted using Dako fluorescent mounting medium. Fluorescent images were captured by a Zeiss inverted microscope under a 63x numeric aperture oil-immersion lens (Carl Zeiss). The size of spermatozoa heads was calculated with ImageJ software.

## Spermatogenic cell fraction purification

Spermatogenic cell fraction purification was performed as previously described (*Shiota et al., 2018*).

## ATAC-seq and data analysis

### Library preparation

Frozen nuclei from four biologically independent purified spermatid fractions were centrifuged at $3000 \times g$ for 5 minutes and then washed in a 1% BSA-PBS solution. Libraries were then prepared from 100,000 nuclei using the ATAC-seq Kit (Catalog No. 53150) from Active Motif, following the manufacturer's protocol (version B9). After quality control (QC) with Qubit and Fragment Analyzer, libraries were subsequently purified using AMPure XP (Beckman Coulter, Catalog No. A63881) beads with a double size selection of 100–800 bp fragments. After a second round of library QC (Qubit and Fragment Analyzer), we performed a paired-end sequencing on a NextSeq 2000 device using a 200-cycle P2 flow cell.

### Trimming

The raw fastq files were processed by 5' end trimming, keeping 30bp-length fragments, using fastx_trimmer (http://hannonlab.cshl.edu/fastx_toolkit/. accessed 28 February 2022), with options -l 30 -Q33.

### Alignment

The trimmed fastq files were aligned on the UCSC Mus_musculus mm10 genome using the Bowtie2 aligner (*Langmead and Salzberg, 2012*), with options –end-to-end, –no-mixed, –no-discordant.

### Normalization

The BigWig files (.bw) containing normalized integrated aligned read count signals were obtained from by normalizing and smoothing bam files using bamCoverage (from deepTools suite *Ramírez et al., 2014*) with options: –binSize 4 –minMappingQuality 30 –normalizeUsing RPM.

### Heatmap

The normalized ATAC-seq signal was converted into a 10 bp bin matrix of the signal 500b upstream and 1500b downstream protein-coding genes' TSS, using computeMatrix (from deepTools suite *Ramírez et al., 2014*), with options reference-point -R mm10_pc.bed –referencePoint TSS –binSize 10 –beforeRegionStartLength 500 –afterRegionStartLength 1500.

Heatmaps were generated using plotHeatmap (deepTools suite *Ramírez et al., 2014*).

## RNA-seq data processing

Wild-type (WT) and *Atad2* knockout (KO) homozygous males were sacrificed by cervical dislocation on days 20-, 22-, 24-, and 26- postpartum (PP), after which their testes were harvested. RNA was extracted using TRIzol reagent, followed by DNA digestion with DNase I. The resulting RNA samples were used for RNA sequencing. For both WT and *Atad2* KO mice, the biological replicates included three samples for D20, four samples for D22, two samples for D24, and two samples for D26.

The raw sequencing data (FASTQ files) were aligned to the UCSC mm10 mouse genome using the STAR software (version 2.7.11b) (*Dobin et al., 2013*) to generate BAM files. Raw read counts were extracted from the BAM files using HTSeq (version 2.0.5) (*Anders et al., 2015*) with the following parameters: -t exon, -f bam, -r pos, `--stranded=reverse`, -m intersection-strict, and `--nonunique` none.

Normalization of read counts was performed using the DESeq2 R package (version 1.22.2) (*Anders and Huber, 2010*; *Love et al., 2014*). Counts were transformed into DESeq2-normalized values and subsequently log-transformed using the formula log2(1+DESeq2-normalized counts). Differential expression analysis was conducted with the R package 'SARTools' (*Varet et al., 2016*).

Purified RNA from fractionated round spermatids was also sequenced to establish gene transcriptional group quartile shown in *Figure 4*. RNA purification and sequencing were performed as described above.

## Quantification and statistical analysis

Statistical analysis was performed with R and GraphPad Prism 8. The results of fertility tests were analyzed by Student *t*-test. Pairwise Student *t*-test was used to analyze testis weights and spermatozoa counts of same-aged mice, and ratio pairwise Student *t*-test was used for the IVF efficiency results. The results of sperm decompaction test were analyzed by two-way ANOVA, and Tukey's HSD post two-way ANOVA test was used to determine the significance between groups. *P*-values <0.05 were considered statistically significant: *$P<0.05$, **$P<0.01$, ***$P<0.001$.

## Gene Set Enrichment Analysis (GSEA)

Differential expression analysis and the corresponding GSEA analysis (*Mootha et al., 2003*; *Subramanian et al., 2005*) were performed to compare *Atad2* KO versus *Atad2* wild-type samples at days 20, 22, 24, and 26 PP. For the GSEA analysis, we used the Python package 'gseapy' (*Fang et al., 2023*) available at https://www.gsea-msigdb.org/gsea.

The GSEA was carried out on three custom gene sets. The first gene set, referred to as 'sex chromosome-linked genes', corresponds to genes located on the X and Y chromosomes. The remaining two gene sets include genes identified as upregulated or downregulated in the absence of histone H3.3 by Fontaine and colleagues (*Fontaine et al., 2022*). These gene sets were designated as 'post-meiotic H3.3-activated genes' and 'post-meiotic H3.3-repressed genes', respectively.

## Acknowledgements

This work was supported by ANR EpiSperm 4 (ANR-19-CE12- 0014), ANR Episperm 5 (ANR-23-CE12-0028), ANR NME-CoA (ANR-23-CE14-0050) and the Cancer ITMO [Multi-Organization Thematic Institute of the French Alliance for Life Sciences and Health (AVIESAN)] MIC program to SK and SR. AL is a recipient of a 4th-year PhD fellowship by Foundation pour la Recherche Médicale (FRM). High-throughput sequencing was performed at the TGML Platform, supported by grants from Inserm, GIS IBiSA, Aix-Marseille Université. We thank Dr. Alexandra Varga for X-gal staining of testis sections. The authors thank Dr Kelly Matmati, Stroke Program Director, Rochester General Hospital, Connecticut, USA, for her critical reading of the manuscript.

## Additional information

### Funding

| Funder | Grant reference number | Author |
| --- | --- | --- |
| Agence Nationale de la Recherche | ANR EpiSperm 4 (ANR-19-CE12- 0014) | Saadi Khochbin |
| Agence Nationale de la Recherche | ANR Episperm 5 (ANR-23-CE12-0028) | Saadi Khochbin |
| Agence Nationale de la Recherche | ANR NME-CoA (ANR-23-CE14-0050) | Saadi Khochbin |

| Funder | Grant reference number | Author |
| --- | --- | --- |

The funders had no role in study design, data collection and interpretation, or the decision to submit the work for publication.

## Author contributions

Ariadni Liakopoulou, Data curation, Formal analysis, Investigation, Methodology; Fayçal Boussouar, Conceptualization, Supervision, Investigation, Methodology, Writing – original draft, Writing – review and editing; Daniel Perazza, Experimentation and data generation and analyses; Sophie Barral, Methodology; Emeline Lambert, In vitro fertilization assays; Tao Wang, Formal analysis, Methodology; Florent Chuffart, Transcriptomic data analyses; Ekaterina Bourova-Flin, Transcriptomic data analyses; Charlyne Gard, Transcriptomic data generation; Denis Puthier, Transcriptomic data generation; Sophie Rousseaux, Funding acquisition, Transcriptomic data analyses; Christophe Arnoult, In vitro fertilization assays; André Verdel, Experimentation and data generation and analyses; Saadi Khochbin, Conceptualization, Resources, Data curation, Formal analysis, Supervision, Funding acquisition, Validation, Investigation, Visualization, Writing – original draft, Project administration, Writing – review and editing

## Author ORCIDs

Ariadni Liakopoulou (ID) https://orcid.org/0009-0005-8964-9205
Fayçal Boussouar (ID) https://orcid.org/0000-0003-4529-4893
Daniel Perazza (ID) https://orcid.org/0000-0002-7703-4984
Sophie Barral (ID) https://orcid.org/0000-0001-9915-6096
Emeline Lambert (ID) https://orcid.org/0009-0005-6909-5831
Tao Wang (ID) https://orcid.org/0000-0002-5178-6969
Florent Chuffart (ID) https://orcid.org/0000-0001-6064-5183
Ekaterina Bourova-Flin (ID) https://orcid.org/0000-0002-5673-9045
Denis Puthier (ID) https://orcid.org/0000-0002-7240-5280
Sophie Rousseaux (ID) https://orcid.org/0000-0001-5246-5350
Christophe Arnoult (ID) https://orcid.org/0000-0002-3753-5901
André Verdel (ID) https://orcid.org/0000-0001-6048-3794
Saadi Khochbin (ID) https://orcid.org/0000-0002-0455-0857

Reviewer #1 (Public review): https://doi.org/10.7554/eLife.107582.3.sa1
Reviewer #2 (Public review): https://doi.org/10.7554/eLife.107582.3.sa2
Reviewer #3 (Public review): https://doi.org/10.7554/eLife.107582.3.sa3
Author response https://doi.org/10.7554/eLife.107582.3.sa4

# Additional files

## Supplementary files

Supplementary file 1. Source data for *Figure 2A*, *Figure 7B*, *Figure 8* and *Figure 6—figure supplement 2*.

MDAR checklist

## Data availability

(1) The transcriptomic data generated in this study, corresponding to testes collected at different days post-partum, have been deposited on GEO under the accession GSE277943. (2) Transcriptomic data from fractionated round spermatids (three biological replicates, Fig. 4A) were deposited on GEO accession GSE284749. (3) ATAC-seq data generated in this study were deposited on GEO under the accession GSE283879.

The following datasets were generated:

| Author(s) | Year | Dataset title | Dataset URL | Database and Identifier |
|---|---|---|---|---|
| Liakopoulou A, Boussouar F, Perazza D, Barral S, Lambert E, Wang T, Chuffart F, Bourova-Flin E, Gard C, Puthier D, Rousseaux S, Arnoult C, Verdel A, Khochbin S | 2025 | ATAD2 mediates chromatin-bound histone chaperone turnover | https://www.ncbi.nlm.nih.gov/geo/query/acc.cgi?acc=GSE277943 | NCBI Gene Expression Omnibus, GSE277943 |
| Liakopoulou A, Boussouar F, Perazza D, Barral S, Lambert E, Wang T, Chuffart F, Bourova-Flin E, Gard C, Puthier D, Rousseaux S, Arnoult C, Verdel A, Khochbin S | 2025 | ATAD2 mediates chromatin-bound histone chaperone turnover | https://www.ncbi.nlm.nih.gov/geo/query/acc.cgi?acc=GSE284749 | NCBI Gene Expression Omnibus, GSE284749 |
| Liakopoulou A, Boussouar F, Perazza D, Barral S, Lambert E, Wang T, Chuffart F, Bourova-Flin E, Gard C, Puthier D, Rousseaux S, Arnoult C, Verdel A, Khochbin S | 2025 | ATAD2 mediates chromatin-bound histone chaperone turnover | https://www.ncbi.nlm.nih.gov/geo/query/acc.cgi?acc=GSE283879 | NCBI Gene Expression Omnibus, GSE283879 |

The following previously published dataset was used:

| Author(s) | Year | Dataset title | Dataset URL | Database and Identifier |
|---|---|---|---|---|
| Fontaine E, Papin C, Martinez G, Le Gras S, Abi Nahed R, Héry P, Buchou T, Ouararhni K, Favier B, Gautier T, Sabir JSM, Gérard M, Bednar J, Arnoult C, Dimitrov S, Hamiche A | 2022 | Dual role of histone variant H3.3B in spermatogenesis: positive regulation of piRNA transcription and implication in X-chromosome inactivation | https://www.ncbi.nlm.nih.gov/geo/query/acc.cgi?acc=GSE116373 | NCBI Gene Expression Omnibus, GSE116373 |

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
