## [Editor Report · eLife Assessment]

This **valuable** study explores the role of the chromatin regulator ATAD2 in mouse spermatogenesis. The data **convincingly** demonstrate that ATAD2 is essential for proper chromatin remodeling in haploid spermatids, influencing gene accessibility, H3.3-mediated transcription, and histone eviction. Using Atad2 knockout (KO) mice, the authors link ATAD2 to the DNA-replication-independent incorporation of sperm-specific proteins like protamines and histone H3.3. Although the findings highlight chromatin abnormalities and impaired in vitro fertilization in KO mice, natural fertility remains unaffected, suggesting possible in vivo compensatory mechanisms. Future experiments will be needed to tease out the precise molecular role of ATAD2 in spermatogenesis. This work will be of interest to the epigenetics and developmental fields.

---

## [Referee Report · Reviewer #1 (Public review)]

Summary:

The authors analyzed the expression of ATAD2 protein in post-meiotic stages and characterized the localization of various testis-specific proteins in the testis of the Atad2 knockout (KO). By cytological analysis as well as the ATAC sequencing, the study showed that increased levels of HIRA histone chaperone, accumulation of histone H3.3 on post-meiotic nuclei, defective chromatin accessibility and also delayed deposition of protamines. Sperm from the Atad2 KO mice reduces the success of in vitro fertilization. The work was performed well, and most of the results are convincing. However, this manuscript does not suggest a molecular mechanism for how ATAD2 promotes the formation of testis-specific chromatin.

Strengths:

The paper describes the role of ATAD2 AAA+ ATPase in the proper localization of sperm-specific chromatin proteins such as protamine, suggesting the importance of the DNA replication-independent histone exchanges with the HIRA-histone H3.3 axis.

Weaknesses:

The work was performed well, and most of the results are convincing. However, this manuscript does not suggest a molecular mechanism for how ATAD2 promotes the formation of testis-specific chromatin.

---

## [Referee Report · Reviewer #2 (Public review)]

Summary:

This manuscript by Liakopoulou et al. presents a comprehensive investigation into the role of ATAD2 in regulating chromatin dynamics during spermatogenesis. The authors elegantly demonstrate that ATAD2, via its control of histone chaperone HIRA turnover, ensures proper H3.3 localization, chromatin accessibility, and histone-to-protamine transition in post-meiotic male germ cells. Using a new well-characterized Atad2 KO mouse model, they show that ATAD2 deficiency disrupts HIRA dynamics, leading to aberrant H3.3 deposition, impaired transcriptional regulation, delayed protamine assembly, and defective sperm genome compaction. The study bridges ATAD2's conserved functions in embryonic stem cells and cancer to spermatogenesis, revealing a novel layer of epigenetic regulation critical for male fertility.

Strengths:

The MS first demonstration of ATAD2's essential role in spermatogenesis, linking its expression in haploid spermatids to histone chaperone regulation by connecting ATAD2-dependent chromatin dynamics to gene accessibility (ATAC-seq), H3.3-mediated transcription, and histone eviction. Interestingly and surprisingly, sperm chromatin defects in Atad2 KO mice impair only in vitro fertilization but not natural fertility, suggesting unknown compensatory mechanisms in vivo.

Weaknesses:

The MS is robust and there are not big weaknesses

The authors have addressed all the queries successfully.

---

## [Referee Report · Reviewer #3 (Public review)]

Summary:

The authors generated knockout mice for Atad2, a conserved bromodomain-containing factor expressed during spermatogenesis. In Atad2 KO mice, HIRA, a chaperone for histone variant H3.3, was upregulated in round spermatids, accompanied by an apparent increase in H3.3 levels. Furthermore, the sequential incorporation and removal of TH2B and PRM1 during spermiogenesis were partially disrupted in the absence of ATAD2, possibly due to delayed histone removal. Despite these abnormalities, Atad2 KO male mice were able to produce offspring normally.

Strengths:

The manuscript addresses the biological role of ATAD2 in spermatogenesis using a knockout mouse model, providing a valuable in vivo framework to study chromatin regulation during male germ cell development. The observed redistribution of H3.3 in round spermatids is clearly presented and suggests a previously unappreciated role of ATAD2 in histone variant dynamics. The authors also document defects in the sequential incorporation and removal of TH2B and PRM1 during spermiogenesis, providing phenotypic insight into chromatin transitions in late spermatogenic stages. Overall, the study presents a solid foundation for further mechanistic investigation into ATAD2 function.

Weaknesses:

While the manuscript reports the gross phenotype of Atad2 KO mice, the findings remain largely superficial and do not convincingly demonstrate how ATAD2 deficiency affects chromatin.

---

## [Author Response]

The following is the authors’ response to the original reviews.

**Public Reviews:**

**Reviewer #1 (Public review):**
Summary:The authors analyzed the expression of ATAD2 protein in post-meiotic stages and characterized the localization of various testis-specific proteins in the testis of the Atad2 knockout (KO). By cytological analysis as well as the ATAC sequencing, the study showed that increased levels of HIRA histone chaperone, accumulation of histone H3.3 on post-meiotic nuclei, defective chromatin accessibility and also delayed deposition of protamines. Sperm from the Atad2 KO mice reduces the success of in vitro fertilization. The work was performed well, and most of the results are convincing. However, this manuscript does not suggest a molecular mechanism for how ATAD2 promotes the formation of testis-specific chromatin.

We would like to take this opportunity to highlight that the present study builds on our previously published work, which examined the function of ATAD2 in both yeast *S. pombe* and mouse embryonic stem (ES) cells (Wang et al., 2021). In yeast, using genetic analysis we showed that inactivation of HIRA rescues defective cell growth caused by the absence of ATAD2. This rescue could also be achieved by reducing histone dosage, indicating that the toxicity depends on histone over-dosage, and that HIRA toxicity, in the absence of ATAD2, is linked to this imbalance.

Furthermore, HIRA ChIP-seq performed in mouse ES cells revealed increased nucleosome-bound HIRA, particularly around transcription start sites (TSS) of active genes, along with the appearance of HIRA-bound nucleosomes within normally nucleosome-free regions (NFRs). These findings pointed to ATAD2 as a major factor responsible for unloading HIRA from nucleosomes. This unloading function may also apply to other histone chaperones, such as FACT (see Wang et al., 2021, Fig. 4C).

In the present study, our investigations converge on the same ATAD2 function in the context of a physiologically integrated mammalian system—spermatogenesis. Indeed, in the absence of ATAD2, we observed H3.3 accumulation and enhanced H3.3-mediated gene expression. Consistent with this functional model of ATAD2— unloading chaperones from histone- and non-histone-bound chromatin—we also observed defects in histone-toprotamine replacement.

Together, the results presented here and in Wang et al. (2021) reveal an underappreciated regulatory layer of histone chaperone activity. Previously, histone chaperones were primarily understood as factors that load histones. Our findings demonstrate that we must also consider a previously unrecognized regulatory mechanism that controls assembled histone-bound chaperones. This key point was clearly captured and emphasized by Reviewer #2 (see below).

Strengths:The paper describes the role of ATAD2 AAA+ ATPase in the proper localization of sperm-specific chromatin proteins such as protamine, suggesting the importance of the DNA replication-independent histone exchanges with the HIRA-histone H3.3 axis.Weaknesses:(1) Some results lack quantification.

We will consider all the data and add appropriate quantifications where necessary.

(2) The work was performed well, and most of the results are convincing. However, this manuscript does not suggest a molecular mechanism for how ATAD2 promotes the formation of testis-specific chromatin.

Please see our comments above.

**Reviewer #2 (Public review):**
Summary:This manuscript by Liakopoulou et al. presents a comprehensive investigation into the role of ATAD2 in regulating chromatin dynamics during spermatogenesis. The authors elegantly demonstrate that ATAD2, via its control of histone chaperone HIRA turnover, ensures proper H3.3 localization, chromatin accessibility, and histone-toprotamine transition in post-meiotic male germ cells. Using a new well-characterized Atad2 KO mouse model, they show that ATAD2 deficiency disrupts HIRA dynamics, leading to aberrant H3.3 deposition, impaired transcriptional regulation, delayed protamine assembly, and defective sperm genome compaction. The study bridges ATAD2's conserved functions in embryonic stem cells and cancer to spermatogenesis, revealing a novel layer of epigenetic regulation critical for male fertility.Strengths:The MS first demonstration of ATAD2's essential role in spermatogenesis, linking its expression in haploid spermatids to histone chaperone regulation by connecting ATAD2-dependent chromatin dynamics to gene accessibility (ATAC-seq), H3.3-mediated transcription, and histone eviction. Interestingly and surprisingly, sperm chromatin defects in Atad2 KO mice impair only in vitro fertilization but not natural fertility, suggesting unknown compensatory mechanisms in vivo.Weaknesses:The MS is robust and there are not big weaknesses
**Reviewer #3 (Public review):**
Summary:The authors generated knockout mice for Atad2, a conserved bromodomain-containing factor expressed during spermatogenesis. In Atad2 KO mice, HIRA, a chaperone for histone variant H3.3, was upregulated in round spermatids, accompanied by an apparent increase in H3.3 levels. Furthermore, the sequential incorporation and removal of TH2B and PRM1 during spermiogenesis were partially disrupted in the absence of ATAD2, possibly due to delayed histone removal. Despite these abnormalities, Atad2 KO male mice were able to produce offspring normally.Strengths:The manuscript addresses the biological role of ATAD2 in spermatogenesis using a knockout mouse model, providing a valuable in vivo framework to study chromatin regulation during male germ cell development. The observed redistribution of H3.3 in round spermatids is clearly presented and suggests a previously unappreciated role of ATAD2 in histone variant dynamics. The authors also document defects in the sequential incorporation and removal of TH2B and PRM1 during spermiogenesis, providing phenotypic insight into chromatin transitions in late spermatogenic stages. Overall, the study presents a solid foundation for further mechanistic investigation into ATAD2 function.Weaknesses:While the manuscript reports the gross phenotype of Atad2 KO mice, the findings remain largely superficial and do not convincingly demonstrate how ATAD2 deficiency affects chromatin dynamics. Moreover, the phenotype appears too mild to elucidate the functional significance of ATAD2 during spermatogenesis.

We respectfully disagree with the statement that our findings are largely superficial. Based on our investigations of this factor over the years, it has become evident that ATAD2 functions as an auxiliary factor that facilitates mechanisms controlling chromatin dynamics (see, for example, Morozumi et al., 2015). These mechanisms can still occur in the absence of ATAD2, but with reduced efficiency, which explains the mild phenotype we observed.

This function, while not essential, is nonetheless an integral part of the cell’s molecular biology and should be studied and brought to the attention of the broader biological community, just as we study essential factors. Unfortunately, the field has tended to focus primarily on core functional actors, often overlooking auxiliary factors. As a result, our decade-long investigations into the subtle yet important roles of ATAD2 have repeatedly been met with skepticism regarding its functional significance, which has in turn influenced editorial decisions.

We chose eLife as the venue for this work specifically to avoid such editorial barriers and to emphasize that facilitators of essential functions do exist. They deserve to be investigated, and the underlying molecular regulatory mechanisms must be understood.

(1) Figures 4-5: The analyses of differential gene expression and chromatin organization should be more comprehensive. First, Venn diagrams comparing the sets of significantly differentially expressed genes between this study and previous work should be shown for each developmental stage. Second, given the established role of H3.3 in MSCI, the effect of Atad2 knockout on sex chromosome gene expression should be analyzed. Third, integrated analysis of RNA-seq and ATAC-seq data is needed to evaluate how ATAD2 loss affects gene expression. Finally, H3.3 ChIP-seq should be performed to directly assess changes in H3.3 distribution following Atad2 knockout.

(1) In the revised version, we will include Venn diagrams to illustrate the overlap in significantly differentially expressed genes between this study and previous work. However, we believe that the GSEAs presented here provide stronger evidence, as they indicate the statistical significance of this overlap (p-values). In our case, we observed p-value < 0.01 (**) and p < 0.001 (*******).

(2) Sex chromosome gene expression was analyzed and is presented in Fig. 5C.

(3) The effect of ATAD2 loss on gene expression is shown in Fig. 4A, B, and C as histograms, with statistical significance indicated in the middle panels.

(4) Although mapping H3.3 incorporation across the genome in wild-type and Atad2 KO cells would have been informative, the available anti-H3.3 antibody did not work for ChIP-seq, at least in our hands. The authors of Fontaine et al., 2022, who studied H3.3 during spermatogenesis in mice, must have encountered the same problem, since they tagged the endogenous H3.3 gene to perform their ChIP experiments.

(2) Figure 3: The altered distribution of H3.3 is compelling. This raises the possibility that histone marks associated with H3.3 may also be affected, although this has not been investigated. It would therefore be important to examine the distribution of histone modifications typically associated with H3.3. If any alterations are observed, ChIP-seq analyses should be performed to explore them further.

Based on our understanding of ATAD2’s function—specifically its role in releasing chromatin-bound HIRA—in the absence of ATAD2 the residence time of both HIRA and H3.3 on chromatin increases. This results in the detection of H3.3 not only on sex chromosomes but across the genome. Our data provide clear evidence of this phenomenon. The reviewer is correct in suggesting that the accumulated H3.3 would carry H3.3-associated histone PTMs; however, we are unsure what additional insights could be gained by further demonstrating this point.

(3) Figure 7: While the authors suggest that pre-PRM2 processing is impaired in Atad2 KO, no direct evidence is provided. It is essential to conduct acid-urea polyacrylamide gel electrophoresis (AU-PAGE) followed by western blotting, or a comparable experiment, to substantiate this claim.

Figure 7 does not suggest that pre-PRM2 processing is affected in Atad2 KO; rather, this figure—particularly Fig. 7B—specifically demonstrates that pre-PRM2 processing is impaired, as shown using an antibody that recognizes the processed portion of pre-PRM2. ELISA was used to provide a more quantitative assessment; however, in the revised manuscript we will also include a western blot image.

(4) HIRA and ATAD2: Does the upregulation of HIRA fully account for the phenotypes observed in Atad2 KO? If so, would overexpression of HIRA alone be sufficient to phenocopy the Atad2 KO phenotype? Alternatively, would partial reduction of HIRA (e.g., through heterozygous deletion) in the Atad2 KO background be sufficient to rescue the phenotype?

These are interesting experiments that require the creation of appropriate mouse models, which are not currently available.

(5) The mechanism by which ATAD2 regulates HIRA turnover on chromatin and the deposition of H3.3 remains unclear from the manuscript and warrants further investigation.

The Reviewer is absolutely correct. In addition to the points addressed in response to Reviewer #1’s general comments (see above), it would indeed have been very interesting to test the segregase activity of ATAD2 (likely driven by its AAA ATPase activity) through in vitro experiments using the Xenopus egg extract system described by Tagami et al., 2004. This system can be applied both in the presence and absence (via immunodepletion) of ATAD2 and would also allow the use of ATAD2 mutants, particularly those with inactive AAA ATPase or bromodomains. However, such experiments go well beyond the scope of this study, which focuses on the role of ATAD2 in chromatin dynamics during spermatogenesis.

References:

(1) Wang T, Perazza D, Boussouar F, Cattaneo M, Bougdour A, Chuffart F, Barral S, Vargas A, Liakopoulou A, Puthier D, Bargier L, Morozumi Y, Jamshidikia M, Garcia-Saez I, Petosa C, Rousseaux S, Verdel A, Khochbin S. ATAD2 controls chromatin-bound HIRA turnover. Life Sci Alliance. 2021 Sep 27;4(12):e202101151. doi: 10.26508/lsa.202101151. PMID: 34580178; PMCID: PMC8500222.

(2) Morozumi Y, Boussouar F, Tan M, Chaikuad A, Jamshidikia M, Colak G, He H, Nie L, Petosa C, de Dieuleveult M, Curtet S, Vitte AL, Rabatel C, Debernardi A, Cosset FL, Verhoeyen E, Emadali A, Schweifer N, Gianni D, Gut M, Guardiola P, Rousseaux S, Gérard M, Knapp S, Zhao Y, Khochbin S. Atad2 is a generalist facilitator of chromatin dynamics in embryonic stem cells. J Mol Cell Biol. 2016 Aug;8(4):349-62. doi: 10.1093/jmcb/mjv060. Epub 2015 Oct 12. PMID: 26459632; PMCID: PMC4991664.

(3) Fontaine E, Papin C, Martinez G, Le Gras S, Nahed RA, Héry P, Buchou T, Ouararhni K, Favier B, Gautier T, Sabir JSM, Gerard M, Bednar J, Arnoult C, Dimitrov S, Hamiche A. Dual role of histone variant H3.3B in spermatogenesis: positive regulation of piRNA transcription and implication in X-chromosome inactivation. Nucleic Acids Res. 2022 Jul 22;50(13):7350-7366. doi: 10.1093/nar/gkac541. PMID: 35766398; PMCID: PMC9303386.

(4) Tagami H, Ray-Gallet D, Almouzni G, Nakatani Y. Histone H3.1 and H3.3 complexes mediate nucleosome assembly pathways dependent or independent of DNA synthesis. Cell. 2004 Jan 9;116(1):51-61. doi: 10.1016/s0092-8674(03)01064-x. PMID: 14718166.

**Recommendations for the authors:**

**Reviewing Editor Comments:**
I note that the reviewers had mixed opinions about the strength of the evidence in the manuscript. A revision that addresses these points would be welcome.
**Reviewer #1 (Recommendations for the authors):**
Major points:(1) No line numbers: It is hard to point out the issues.

The revised version harbors line numbers.

(2) Given the results shown in Figure 3 and Figure 4, it is nice to show the chromosomal localization of histone H3.3 in spermatocytes or post-meiotic cells by Chromatin-immunoprecipitation sequencing (ChIP-seq).

Although mapping H3.3 incorporation across the genome in wild-type and Atad2 KO cells would have been informative, the available anti-H3.3 antibody did not work for ChIP-seq in our hands. In fact, this antibody is not well regarded for ChIP-seq. For example, Fontaine et al. (2022), who investigated H3.3 during spermatogenesis in mice, circumvented this issue by tagging the endogenous H3.3 genes for their ChIP experiments.

(3) Figure 7B and 8: Why the authors used ELISA for the protein quantification. At least, western blotting should be shown.

ELISA is a more quantitative method than traditional immunoblotting. Nevertheless, as requested by the reviewer, we have now included a corresponding western blot in Fig. S3.

(4) For readers, please add a schematic pathway of histone-protamine replacement in sperm formation in Fig.1 and it would be nice to have a model figure, which contains the authors' idea in the last figure.

As requested by this reviewer, we have now included a schematic model in Figure 9 to summarize the main conclusions of our work.

Minor points:(1) Page 2, the second paragraph, "pre-PRM2: Please explain more about pre-PRM2 and/or PRM2 as well as PRM1 (Figure 6).

More detailed descriptions of PRM2 processing are now given in this paragraph.

(2) Page 3, bottom paragraph, line 1: "KO" should be "knockout (KO)".

Done.

(3) Page 4, second paragraph bottom: Please explain more about the protein structure of germ-line-specific ATAD2S: how it is different from ATAD2L. Germ-line specific means it is also expressed in ovary?

As Atad2 is predominantly expressed in embryonic stem cells and in spermatogenic cells, we replaced all through the text germ-line specific by more appropriate terms.

(4) Figure 1C, western blotting: Wild-type testis extracts, both ATAD2L and -S are present. Does this mean that ATADS2L is expressed in both germ line as well as supporting cells. Please clarify this and, if possible, show the western blotting of spermatids well as spermatocytes.

Figure 1D shows sections of seminiferous tubules from Atad2 KO mice, in which lacZ expression is driven by the endogenous Atad2 promoter. The results indicate that Atad2 is expressed mainly in post-meiotic cells. Most labeled cells are located near the lumen, whereas the supporting Sertoli cells remain unlabeled. Sertoli cells, which are anchored to the basal lamina, span the entire thickness of the germinal epithelium from the basal lamina to the lumen. Their nuclei, however, are usually positioned closer to the basal membrane. Thus, the observed lacZ expression pattern argues against substantial Atad2 expression in Sertoli cells.

(5) Figure 1C: Please explain a bit more about the reduction of ATAD2 proteins in heterozygous mice.

Done

(6) Figure 1C: Genotypes of the mice should be shown in the legend.

Done

(7) Figure 1D: Please add a more magnified image of the sections to see the staining pattern in the seminiferous tubules.

The magnification does not bring more information since we lose the structure of cells within tubules due the nature of treatment of the sections for X-gal staining. Please see comments to question 1C to reviewer 2

(8) Page 5, first paragraph, line 2, histone dosage: What do the authors meant by the histone dosage? Please explain more or use more appropriate word.

"Histone dosage" refers to the amount or relative abundance of histone proteins in a cell.

(9) Figure 2A: Figure 2A: Given the result in Figure 1C, it is interesting to check the amount of HIRA in Atad2 heterozygous mice.

In Atad2 heterozygous mice, we would expect an increase in HIRA, but only to about half the level seen in the Atad2 homozygous knockout shown in Figure 2A, which is relatively modest. Therefore, we doubt that detecting such a small change—approximately half of that in Figure 2A—would yield clear or definitive results.

(10) Figure 2A, legend (n=5): What does this "n" mean? The extract of testes from "5" male mice like Figure 2B. Or 5 independent experiments. If the latter is true, it is important to share the other results in the Supplements.

“n” refers to five WT and five Atad2 KO males. The legend has been clarified as suggested by the reviewer.

(11) Figure 2A, legend, line 2, Atad2: This should be italicized.

Done

(12) Figure 2B: Please show the quantification of amounts of HIRA protein like Fig. 2A.

As indicated in the legend, what is shown is a pool of testes from 3 individuals per genotype.

(13) Figure 2B shows an increased level of HIRA in Atad2 KO testis. This suggests the role of ATAD2 in the protein degradation of HIRA. This possibility should be mentioned or tested since ATAD2 is an AAA+ ATPase.

The extensive literature on ATAD2 provides no indication that it is involved in protein degradation. In our early work on ATAD2 in the 2000s, we hypothesized that, as a member of the AAA ATPase family, ATAD2 might associate with the 19S proteasome subunit (through multimerization with the other AAA ATPase member of this regulatory subunit). However, both our published pilot studies (Caron et al., PMID: 20581866) and subsequent unpublished work ruled out this possibility. Instead, since the amount of nucleosome-bound HIRA increases in the absence of ATAD2, we propose that chromatin-bound HIRA is more stable than soluble HIRA once it has been released from chromatin by ATAD2.

(14) Page 6, second paragraph, line 5, ko: KO should be capitalized.

Done

(15) Page 6, second paragraph, line 2 from the bottom, chromatin dynamics: Throughout the text, the authors used "chromatin dynamics". However, all the authors analyzed in the current study is the localization of chromatin protein. So, it is much easier to explain the results by using "chromatin status," etc. In this context, "accessibility" is better.

We changed the term “chromatin dynamics” into a more precise term according to the context used all through the text.

(16) Figure 3: Please provide the quantification of signals of histone H3.3 in a nucleus or nuclear cytoplasm.

This request is not clear to us since we do not observe any H3.3 signal in the cytoplasm.

(17) Figure 3: As the control of specificity in post-meiotic cells, please show the image and quantification of the H3.3 signals in spermatocyte, for example.

This request is not clear to us. What specificity is meant?

(18) Figure 3, bottom panels: Please show what the white lines indicate?

The white lines indicate the limit of cell nucleus and estimated by Hoechst staining. This is now indicated in the legend of the figure.

(19) Figure 4A: Please explain more about what kind of data is here. Is this wild-type and/or Atad2 KO? The label of the Y-axis should be "mean expression level". What is the standard deviation (SD) here on the X-axis. Moreover, there is only one red open circle, but the number of this class is 5611. All 5611 genes in this group show NO expression. Please explain more.

The plot displays the mean expression levels (y-axis, labeled as "mean expression level") versus the corresponding standard deviations (x-axis), both calculated from three independent biological replicates of isolated round spermatids (Atad2 wild-type and Atad2 KO). The standard deviation reflects the variability of gene expression across biological replicates. Genes were grouped into four categories (grp1: blue, grp2: cyan, grp3: green, grp4: orange) according to the quartile of their mean expression. For grp4, all genes have no detectable expression, resulting in a mean expression of zero and a standard deviation of zero; consequently, the 5611 genes in this group are represented by a single overlapping point (red open circle) at the origin.

(20) Figure 4C: If possible, it would be better to have a statistical comparison between wild-type and the KO.

The mean profiles are displayed together with their variability (± 2 s.e.m.) across the four replicates for both ATAD2 WT (blue) and ATAD2 KO (red). For groups 1, 2, and 3, the envelopes of the curves remain clearly separated around the peak, indicating a consistent difference in signal between the two conditions. In contrast, group 4 does not present a strong signal and, accordingly, no marked difference is observed between WT and KO in this group.

(21) Figure 5, GSEA panels: Please explain more about what the GSEA is in the legend. The legend has been updated as follows:

(A) Expression profiles of post-meiotic H3.3-activated genes. The heatmap (left panel) displays the normalized expression levels of genes identified by Fontaine and colleagues as upregulated in the absence of histone H3.3 (Fontaine et al. 2022) for Atad2 WT (WT) and Atad2 KO (KO) samples at days 20, 22, 24, and 26 PP (D20 to D26). The colour scale represents the z-score of log-transformed DESeq2-normalized counts. The middle panel box plots display, pooled, normalized expression levels, aggregated across replicates and genes, for each condition (WT and KO) and each time point (D20 to D26). Statistical significance between WT and KO conditions was determined using a two-sided t-test, with p-values indicated as follows: * for p-value<0.05, ** for p-value<0.01 and *** for p-value<0.001. The right panel shows the results of gene set enrichment analysis (GSEA), which assesses whether predefined groups of genes show statistically significant differences between conditions. Here, the post-meiotic H3.3-activated genes set, identified by Fontaine et al. (2022), is significantly enriched in Atad2 KO compared with WT samples at day 26 (p < 0.05, FDR < 0.25). Coloured vertical bars indicate the “leading edge” genes (i.e., those contributing most to the enrichment signal), located before the point of maximum enrichment score. (B) As shown in (A) but for the "post-meiotic H3.3-repressed genes" gene set. (C) As shown in (A) but for the " sex chromosome-linked genes " gene set.

(22) Figure 6. In the KO, the number of green cells is more than red and yellow cells, suggesting the delayed maturation of green (TH2B-positive) cells. It is essential to count the number of each cell and show the quantification.

The green cells correspond to those expressing TH2B but lacking transition proteins (TP) and protamine 1 (Prm1), indicating that they are at earlier stages than elongating–condensing spermatids. Counting these green cells simply reflects the ratio of elongating/condensing spermatids to earlier-stage cells, which varies depending on the field examined. The key point in this experiment is that in wild-type mice, only red cells (elongating/condensing spermatids) and green cells (earlier stages) are observed. By contrast, in Atad2 KO testes, a significant proportion of yellow cells appears, which are never seen in wild-type tissue. The crucial metric here is the percentage of yellow cells relative to the total number of elongating/condensing spermatids (red cells). In wild-type testes, this value is consistently 0%, whereas in Atad2 KO testes it always ranges between 50% and 100% across all fields containing substantial numbers of elongating/condensing spermatids.

(23) Figure 8A: Please show the images of sperm (heads) in the KO mice with or without decompaction.

The requested image is now displayed in Figure S5.

(24) Figure 8C: In the legend, it says n=5. However, there are more than 5 plots on the graph. Please explain the experiment more in detail.

The experiment is now better explained in the legend of this Figure.

**Reviewer #2 (Recommendations for the authors):**
While the study is rigorous and well performed, the following minor points could be addressed to strengthen the manuscript:Figure 1C should indicate each of the different types of cells present in the sections. It would be of interest to show specifically the different post-meiotic germ cells.

With this type of sample preparation, it is difficult to precisely distinguish the different cell types within the sections. Nevertheless, the staining pattern strongly indicates that most of the intensely stained cells are post-meiotic, situated near the tubule lumens and extending roughly halfway toward the basal membrane.

In the absence of functional ATAD2, the accumulation of HIRA primarily occurs in round spermatids (Fig. 2B). If technically possible, it would be of great interest to show this by IHC of testis section.

Unfortunately, our antibody did not satisfactorily work in IHC.

The increased of H3.3 signal in Atad2 KO spermatids (Fig. 3) is interpreted because of a reduced turnover. However, alternative explanations (e.g., H3.3 misincorporation or altered chaperone affinity) should not be ruled out.

The referee is correct that alternative explanations are possible. However, based on our previous work (Wang et al., 2021; PMID: 34580178), we demonstrated that in the absence of ATAD2, there is reduced turnover of HIRAbound nucleosomes, as well as reduced nucleosome turnover, evidenced by the appearance of nucleosomes in regions that are normally nucleosome-free at active gene TSSs. We have no evidence supporting any other alternative hypothesis.

In the MS the reduced accessibility at active genes (Fig. 4) is attributed to H3.3 overloading. However, global changes in histone acetylation (e.g., H4K5ac) or other remodelers in KO cells could be also consider.

In fact, we meant that histone overloading could be responsible for the altered accessibility. This has been clearly demonstrated in case of *S. cerevisiae* in the absence of Yta7 (*S. cerevisiae*’ ATAD2) (PMID: 25406467).

In relation with the sperm compaction assay (Fig. 8A), the DTT/heparin/Triton protocol may not fully reflect physiological decompaction. This could be validated with alternative methods (e.g., MNase sensitivity).

The referee is right, but since this is a subtle effect as it can be judged by normal fertility, we doubt that milder approaches could reveal significant differences between wildtype and Atad2 KO sperms.

It is surprising that despite the observed alterations in the genome organization of the sperm, the natural fertility of the KO mice is not affected (Fig. 8C). This warrants deeper discussion: Is functional compensation occurring (e.g., by p97/VCP)? Analysis of epididymal sperm maturation or uterine environment could provide insights.

As detailed in the Discussion section, this work, together with our previous study (Wang et al., 2021; PMID: 34580178), highlights an overlooked level of regulation in histone chaperone activity: the release of chromatinbound factors following their interaction with chromatin. This is an energy-dependent process, driven by ATP and the associated ATPase activity of these factors. Such activity could be mediated by various proteins, such as p97/VCP or DNAJC9–HSP70, as discussed in the manuscript, or by yet unidentified factors. However, most of these mechanisms are likely to occur during the extensive histone-to-histone variant exchanges of meiosis and post-meiotic stages. To the best of our knowledge, epididymal sperm maturation and the uterine environment do not involve substantial histone-to-histone or histone-to-protamine exchanges.

The authors showed that MSCI genes present an enhancement of repression in the absence of ATAD2 by enhancing H3.3 function. It would be also of interest to analyze the behavior of the Sex body during its silencing (zygotene to pachytene) by looking at different markers (i.e., gamma-H2AX phosphorylation, Ubiquitylation etc).

The referee is correct that this is an interesting question. Accordingly, in our future work, we plan to examine the sex body in more detail during its silencing, using a variety of relevant markers, including those suggested by the reviewer. However, we believe that such investigations fall outside the scope of the present study, which focuses on the molecular relationship between ATAD2 and H3.3, rather than on the role of H3.3 in regulating sex body transcription. For a comprehensive analysis of this aspect, studies should primarily focus on the H3.3 mouse models reported by Fontaine and colleagues (PMID: 35766398).

Fig. 6: Co-staining of TH2B/TP1/PRM1 is convincing but would benefit from quantification (% cells with overlapping signals).

The green cells correspond to those expressing TH2B but lacking transition proteins (TP) and protamine 1 (Prm1), indicating that they are at earlier stages than elongating–condensing spermatids. Counting these green cells simply reflects the ratio of elongating/condensing spermatids to earlier-stage cells, which varies depending on the field examined. The key point is that in wild-type mice, only red cells (elongating/condensing spermatids) and green cells (earlier stages) are observed. By contrast, in Atad2 KO testes, a significant proportion of yellow cells appears, which are never seen in wild-type tissue. The crucial metric is the percentage of yellow cells relative to the total number of elongating/condensing spermatids (red cells). In wild-type testes, this value is consistently 0%, whereas in Atad2 KO testes it always ranges between 50% and 100% across all fields containing substantial numbers of elongating/condensing spermatids.